# REANALYSIS OF VARIANCE REDUCED TEMPORAL DIFFERENCE LEARNING

**Tengyu Xu[†], Zhe Wang[†], Yi Zhou[§], Yingbin Liang[†]**
[†] Department of ECE, The Ohio State University ,Columbus, OH 43210, USA
[§] Department of ECE, The University of Utah, Salt Lake City, UT 84112, USA
`xu.3260@osu.edu,wang.10982@osu.edu,yi.zhou@utah.edu,liang.889@osu.edu`

## ABSTRACT

Temporal difference (TD) learning is a popular algorithm for policy evaluation in reinforcement learning, but the vanilla TD can substantially suffer from the inherent optimization variance. A variance reduced TD (VRTD) algorithm was proposed by Korda and La (2015), which applies the variance reduction technique directly to the online TD learning with Markovian samples. In this work, we first point out the technical errors in the analysis of VRTD in Korda and La (2015), and then provide a mathematically solid analysis of the non-asymptotic convergence of VRTD and its variance reduction performance. We show that VRTD is guaranteed to converge to a neighborhood of the fixed-point solution of TD at a linear convergence rate. Furthermore, the variance error (for both i.i.d. and Markovian sampling) and the bias error (for Markovian sampling) of VRTD are significantly reduced by the batch size of variance reduction in comparison to those of vanilla TD. As a result, the overall computational complexity of VRTD to attain a given accurate solution outperforms that of TD under Markov sampling and outperforms that of TD under i.i.d. sampling for a sufficiently small conditional number.

## 1 INTRODUCTION

In reinforcement learning (RL), policy evaluation aims to obtain the expected long-term reward of a given policy and plays an important role in identifying the optimal policy that achieves the maximal cumulative reward over time Bertsekas and Tsitsiklis (1995); Dayan and Watkins (1992); Rummery and Niranjan (1994). The temporal difference (TD) learning algorithm, originally proposed by Sutton (1988), is one of the most widely used policy evaluation methods, which uses the Bellman equation to iteratively bootstrap the estimation process and continually update the value function in an incremental way. In practice, if the state space is large or infinite, function approximation is often used to find an approximate value function efficiently. Theoretically, TD with linear function approximation has been shown to converge to the fixed point solution with i.i.d. samples and Markovian samples in Sutton (1988); Tsitsiklis and Van Roy (1997). The finite sample analysis of TD has also been studied in Bhandari et al. (2018); Dalal et al. (2018a); Cai et al. (2019); Srikant and Ying (2019).

Since each iteration of TD uses one or a mini-batch of samples to estimate the mean of the pseudo-gradient [1], TD learning usually suffers from the inherent *variance*, which substantially degrades the convergence accuracy. Although a diminishing stepsize or very small constant stepsize can reduce the variance Bhandari et al. (2018); Srikant and Ying (2019), they also slow down the convergence significantly.

Two approaches have been proposed to reduce the variance. The first approach is the so-called batch TD, which takes a fixed sample set and transforms the empirical mean square projected Bellman error (MSPBE) into an equivalent convex-concave saddle-point problem Du et al. (2017). Due to the finite-sample nature of such a problem, stochastic variance reduction techniques for conventional optimization can be directly applied here to reduce the variance. In particular, Du et al. (2017) showed that SVRG Johnson and Zhang (2013) and SAGA Defazio et al. (2014) can be applied to improve

---

[1]We call the increment in each iteration of TD as "pseudo-gradient" because such a quantity is not a gradient of any objective function, but the role that it serves in the TD algorithm is analogous to the role of the gradient in the gradient descent algorithm.

the performance of batch TD algorithms, and Peng et al. (2019) proposed two variants of SVRG to further save the computation cost. However, the analysis of batch TD does not take into account the statistical nature of the training samples, which are generated by a MDP. Hence, there is no guarantee of such obtained solutions to be close to the fixed point of TD learning.

The second approach is the so-called TD with centering (CTD) algorithm proposed in Korda and La (2015), which introduces the variance reduction idea to the original TD learning algorithm. For the sake of better reflecting its major feature, we refer to CTD as Variance Reduced TD (VRTD) throughout this paper. Similarly to the SVRG in Johnson and Zhang (2013), VRTD has outer and inner loops. The beginning of each inner-loop (i.e. each epoch) computes a batch of sample pseudo-gradients so that each subsequent inner loop iteration modifies only one sample pseudo-gradient in the batch pseudo-gradients to reduce the variance. The main difference between VRTD and batch TD is that VRTD applies the variance reduction directly to TD learning rather than to a transformed optimization problem in batch TD. Though Korda and La (2015) empirically verified that VRTD has better convergence accuracy than vanilla TD learning, some technical errors in the analysis in Korda and La (2015) have been pointed out in follow up studies Dalal et al. (2018a); Narayanan and Szepesvári (2017). Furthermore, as we discuss in Section 3, the technical proof in Korda and La (2015) regarding the convergence of VRTD also has technical errors so that their results do not correctly characterize the impact of variance reduction on TD learning. Given the recent surge of interest in the finite time analysis of the vanilla TD Dalal et al. (2018a); Bhandari et al. (2018); Dalal et al. (2018b); Srikant and Ying (2019), it becomes imperative to reanalyze the VRTD and accurately understand whether and how variance reduction can help to improve the convergence accuracy over vanilla TD. Towards this end, this paper specifically addresses the following central questions.

- For i.i.d. sampling, it has been shown in Dalal et al. (2018a); Bhandari et al. (2018) that vanilla TD converges only to a neighborhood of the fixed point for a constant stepsize and suffers from a constant error term caused by the variance of the stochastic pseudo-gradient at each iteration. For VRTD, does the variance reduction help to reduce such an error and improve the accuracy of convergence? How does the error depend on the variance reduction parameter, i.e., the batch size for variance reduction?
- For Markovian sampling, it has been shown in Bhandari et al. (2018) that the convergence of vanilla TD further suffers from a bias error due to the correlation among samples in addition to the variance error as in i.i.d. sampling. Does VRTD, which was designed to have reduced variance, also enjoy reduced bias error? If so, how does the bias error depend on the batch size for variance reduction?
- For both i.i.d. and Markovian sampling, to attain an $\epsilon$-accurate solution, what is the overall computational complexity (the total number of computations of pseudo-gradients) of VRTD, and does VRTD have a reduced overall computational complexity compared to TD?

## 1.1 OUR CONTRIBUTIONS

Our main contributions are summarized in Table 1 and are described as follows.

For i.i.d. sampling, we show that a slightly modified version of VRTD (for avoiding bias error) converges linearly to a neighborhood of the fixed point solution for a constant stepsize $\alpha$, with the variance error at the order of $\mathcal{O}(\alpha/M)$, where $M$ is the batch size for variance reduction. To attain an $\epsilon$-accurate solution, the overall computational complexity (i.e., the total number of pseudo-gradient computations) of VRTD outperforms the vanilla TD algorithm Bhandari et al. (2018) if the condition number is small.

For Markovian sampling, we show that VRTD has the same linear convergence and the same variance error reduction over the vanilla TD Dalal et al. (2018a); Bhandari et al. (2018) as i.i.d. sampling. More importantly, the variance reduction in VRTD also attains a substantially reduced bias error at the order of $\mathcal{O}(1/M)$ over the vanilla TD Bhandari et al. (2018), where the bias error is at the order of $\mathcal{O}(\alpha)$. As a result, VRTD outperforms vanilla TD in terms of the total computational complexity by a factor of $\log \frac{1}{\epsilon}$.

At the technical level, our analysis of bias error for Markovian sampling takes a different path from how the existing analysis of TD handles Markovian samples in Bhandari et al. (2018); Wang et al. (2017); Srikant and Ying (2019). Due to the batch average of stochastic pseudo-gradients adopted by VRTD to reduce the variance, the correlation among samples in different epochs is eliminated. Such an analysis explicitly explains why the variance reduction step helps to further reduce the bias error.

Table 1: Comparison of performance of TD and VRTD algorithms.

| | Algorithm | Variance Error | Bias Error | Overall Complexity |
|---|---|---|---|---|
| i.i.d. sample | TD | $\mathcal{O}(\alpha)$ | NA | $\mathcal{O}\left(\frac{1}{\epsilon \lambda_A^2} \log\left(\frac{1}{\epsilon}\right)\right)$ |
| | VRTD | $\mathcal{O}(\alpha/M)$ | NA | $\mathcal{O}\left(\max\left\{\frac{1}{\epsilon}, \frac{1}{\lambda_A^2}\right\} \log\left(\frac{1}{\epsilon}\right)\right)$ |
| Markovian sample | TD | $\mathcal{O}(\alpha)$ | $\mathcal{O}(\alpha)$ | $\mathcal{O}\left(\frac{1}{\epsilon \lambda_A^2} \log^2\left(\frac{1}{\epsilon}\right)\right)$ |
| | VRTD | $\mathcal{O}(\alpha/M)$ | $\mathcal{O}(1/M)$ | $\mathcal{O}\left(\max\left\{\frac{1}{\epsilon}, \frac{1}{\epsilon \lambda_A^2}\right\} \log\left(\frac{1}{\epsilon}\right)\right)$ |

Note: The results on the performance of TD are due to Bhandari et al. (2018) and the results on the performance of VRTD (which are highlighted by the red color) are characterized by this work.

## 1.2 RELATED WORK

**On-policy TD learning and variance reduction.** On-policy TD learning aims to minimize the Mean Squared Bellman Error (MSBE) Sutton (1988) when samples are drawn independently from the stationary distribution of the corresponding MDP. The non-asymptotic convergence under i.i.d. sampling has been established in Dalal et al. (2018a) for TD with linear function approximation and for TD with overparameterized neural network approximation Cai et al. (2019). The convergence of averaged linear SA with constant stepsize has been studied in Lakshminarayanan and Szepesvari (2018). In the Markovian setting, the non-asymptotic convergence has been studied for on-policy TD in Bhandari et al. (2018); Karmakar and Bhatnagar (2016); Wang et al. (2019); Srikant and Ying (2019). Korda and La (2015) proposed a variance reduced CTD algorithm (called VRTD in this paper), which directly applies variance reduction technique to the TD algorithm. The analysis of VRTD provided in Korda and La (2015) has technical errors. The aim of this paper is to provide a technically solid analysis for VRTD to characterize the advantage of variance reduction.

**Variance reduced batch TD learning.** Batch TD Lange et al. (2012) algorithms are generally designed for policy evaluation by solving an optimization problem on a fixed dataset. In Du et al. (2017), the empirical MSPBE is first transformed into a quadratic convex-concave saddle-point optimization problem and variance reduction methods of SVRG Johnson and Zhang (2013) and SAGA Defazio et al. (2014) were then incorporated into a primal-dual batch gradient method. Furthermore, Peng et al. (2019) applied two variants of variance reduction methods to solve the same saddle point problems, and showed that those two methods can save pseudo-gradient computation cost.

We note that due to the extensive research in TD learning, we include here only studies that are highly related to our work, and cannot cover many other interesting topics on TD learning such as asymptotic convergence of TD learning Tadić (2001); Hu and Syed (2019), off-policy TD learning Sutton et al. (2008; 2009); Liu et al. (2015); Wang et al. (2017); Karmakar and Bhatnagar (2017), two time-scale TD algorithms Xu et al. (2019); Dalal et al. (2018b); Yu (2017), fitted TD algorithms Lee and He (2019), SARSA Zou et al. (2019) etc. The idea of the variance reduction algorithm proposed in Korda and La (2015) as well as the analysis techniques that we develop in this paper can potentially be useful for these algorithms.

## 2 PROBLEM FORMULATION AND PRELIMINARIES

### 2.1 ON-POLICY VALUE FUNCTION EVALUATION

We describe the problem of value function evaluation over a Markov decision process (MDP) $(\mathcal{S}, \mathcal{A}, \mathsf{P}, r, \gamma)$, where each component is explained in the sequel. Suppose $\mathcal{S} \subset \mathbb{R}^d$ is a compact state space, and $\mathcal{A}$ is a finite action set. Consider a stationary policy $\pi$, which maps a state $s \in \mathcal{S}$ to the actions in $\mathcal{A}$ via a probability distribution $\pi(\cdot|s)$. At time-step $t$, suppose the process is in some state $s_t \in \mathcal{S}$, and an action $a_t \in \mathcal{A}$ is taken based on the policy $\pi(\cdot|s_t)$. Then the transition kernel $\mathsf{P} = \mathsf{P}(s_{t+1}|s_t, a_t)$ determines the probability of being at state $s_{t+1} \in \mathcal{S}$ in the next time-step, and the reward $r_t = r(s_t, a_t, s_{t+1})$ is received, which is assumed to be bounded by $r_{\max}$. We denote the associated Markov chain by $p(s'|s) = \sum_{a \in \mathcal{A}} p(s'|s, a)\pi(a|s)$, and assume that it is ergodic. Let $\mu_\pi$ be the induced stationary distribution, i.e., $\sum_s p(s'|s)\mu_\pi(s) = \mu_\pi(s')$. We define the value function for a policy $\pi$ as $v^\pi(s) = \mathbb{E}[\sum_{t=0}^{\infty} \gamma^t r(s_t, a_t, s_{t+1})|s_0 = s, \pi]$, where $\gamma \in (0, 1)$ is the discount

factor. Define the Bellman operator $T^\pi$ for any function $\xi(s)$ as $T^\pi \xi(s) := r^\pi(s) + \gamma \mathbb{E}_{s'|s} \xi(s')$, where $r^\pi(s) = \mathbb{E}_{a,s'|s} r(s, a, s')$ is the expected reward of the Markov chain induced by the policy $\pi$. It is known that $v^\pi(s)$ is the unique fixed point of the Bellman operator $T^\pi$, i.e., $v^\pi(s) = T^\pi v^\pi(s)$. In practice, since the MDP is unknown, the value function $v^\pi(s)$ cannot be directly obtained. The goal of policy evaluation is to find the value function $v^\pi(s)$ via sampling the MDP.

## 2.2 TD LEARNING WITH LINEAR FUNCTION APPROXIMATION

In order to find the value function efficiently particularly for large or infinite state space $\mathcal{S}$, we take the standard linear function approximation $\hat{v}(s, \theta) = \phi(s)^\top \theta$ of the value function, where $\phi(s)^\top = [\phi_1(s), \cdots, \phi_d(s)]$ with $\phi_i(s)$ for $i = 1, 2, \cdots d$ denoting the fixed basis feature functions of state $s$, and $\theta \in \mathbb{R}^d$ is a parameter vector. Let $\Phi$ be the $|\mathcal{S}| \times d$ feature matrix (with rows indexed by the state and columns corresponding to components of $\theta$). The linear function approximation can be written in the vector form as $\hat{v}(\theta) = \Phi \theta$. Our goal is to find the fixed-point parameter $\theta^* \in \mathbb{R}^d$ that satisfies $\mathbb{E}_{\mu_\pi} \hat{v}(s, \theta^*) = \mathbb{E}_{\mu_\pi} T^\pi \hat{v}(s, \theta^*)$. The TD learning algorithm performs the following fixed-point iterative update to find such $\theta^*$.

$$\theta_{t+1} = \theta_t + \alpha_t g_{x_t}(\theta_t) = \theta_t + \alpha_t(A_{x_t}\theta_t + b_{x_t}), \tag{1}$$

where $\alpha_t > 0$ is the stepsize, and $A_{x_t}$ and $b_{x_t}$ are specified below. For i.i.d. samples generated from the distribution $\mu_\pi$, we denote the sample as $x_t = (s_t, r_t, s_t')$, and $A_{x_t} = \phi(s_t)(\gamma\phi(s_t') - \phi(s_t))^\top$ and $b_{x_t} = r(s_t)\phi(s_t)$. For Markovian samples generated sequentially from a trajectory, we denote the sample as $x_t = (s_t, r_t, s_{t+1})$, and in this case $A_{x_t} = \phi(s_t)(\gamma\phi(s_{t+1}) - \phi(s_t))^\top$ and $b_{x_t} = r(s_t)\phi(s_t)$. We further define the mean pseudo-gradient $g(\theta) = A\theta + b$ where $A = \mathbb{E}_{\mu_\pi}[\phi(s)(\gamma\phi(s') - \phi(s))^\top]$ and $b = \mathbb{E}_{\mu_\pi}[r(s)\phi(s)]$. We call $g(\theta)$ as pseudo-gradient for convenience due to its analogous role as in the gradient descent algorithm. It has been shown that the iteration in eq. (1) converges to the fix point $\theta^* = -A^{-1}b$ at a sublinear rate $\mathcal{O}(1/t)$ with diminishing stepsize $\alpha_t = \mathcal{O}(1/t)$ using both Markovian and i.i.d. samples Bhandari et al. (2018); Dalal et al. (2018a). Throughout the paper, we make the following standard assumptions Wang et al. (2017); Korda and La (2015); Tsitsiklis and Van Roy (1997); Bhandari et al. (2018).

**Assumption 1** (Problem solvability). *The matrix $A$ is non-singular.*

**Assumption 2** (Bounded feature). $\|\phi(s)\|_2 \leq 1$ *for all $s \in \mathcal{S}$.*

**Assumption 3** (Geometric ergodicity). *The considered MDP is irreducible and aperiodic, and there exist constants $\kappa > 0$ and $\rho \in (0, 1)$ such that*

$$\sup_{s \in \mathcal{S}} d_{TV}(\mathbb{P}(s_t \in \cdot | s_0 = s), \mu_\pi(s)) \leq \kappa\rho^t, \quad \forall t \geq 0,$$

*where $d_{TV}(P, Q)$ denotes the total-variation distance between the probability measures $P$ and $Q$.*

Assumption 1 requires the matrix $A$ to be non-singular so that the optimal parameter $\theta^* = -A^{-1}b$ is well defined. Assumption 2 can be ensured by normalizing the basis functions $\{\phi_i\}_{i=1}^d$. Assumption 3 holds for any time-homogeneous Markov chain with finite state-space and any uniformly ergodic Markov chains with general state space.

## 3 THE VARIANCE REDUCED TD ALGORITHM

In this section, we first introduce the variance-reduced TD (VRTD) algorithm proposed in Korda and La (2015) for Markovian sampling and then discuss the technical errors in the analysis of VRTD in Korda and La (2015).

### 3.1 VRTD ALGORITHM KORDA AND LA (2015)

Since the standard TD learning takes only one sample in each update as can be seen in eq. (1), it typically suffers from a large variance. This motivates the development of the VRTD algorithm in Korda and La (2015) (named as CTD in Korda and La (2015)). VRTD is formally presented in Algorithm 2, and we briefly introduce the idea below. The algorithm runs in a nested fashion with each inner-loop (i.e., each epoch) consists of $M$ updates. At the beginning of the $m$-th epoch, a batch of $M$ samples are acquired and a batch pseudo-gradient $g_m(\tilde{\theta}_{m-1})$ is computed based on these samples as an estimator of the mean pseudo-gradient. Then, each inner-loop update randomly takes

---

**Algorithm 1** Variance Reduced TD with iid samples

**Input:** batch size $M$, learning rate $\alpha$ and initialization $\tilde{\theta}_0$
1: **for** $m = 1, 2, ..., S$ **do**
2:      $\theta_{m,0} = \tilde{\theta}_{m-1}$
3:      Sample a set $B_m$ with $M$ samples indepedently from the distribution $\mu_\pi$
4:      $g_m(\tilde{\theta}_{m-1}) = \frac{1}{M} \sum_{x_i \in B_m} g_{x_i}(\tilde{\theta}_{m-1})$
5:      **for** $t = 0, 1, ..., M - 1$ **do**
6:          Sample $x_{j_{m,t}}$ indepedently from the distribution $\mu_\pi$
7:          $\theta_{m,t+1} = \theta_{m,t} + \alpha\big(g_{x_{j_{m,t}}}(\theta_{m,t})$
8:              $- g_{x_{j_{m,t}}}(\tilde{\theta}_{m-1}) + g_m(\tilde{\theta}_{m-1})\big)$
9:      **end for**
10:     set $\tilde{\theta}_m = \theta_{m,t}$ for randomly chosen $t \in \{1, 2, ..., M\}$
11: **end for**
**Output:** $\tilde{\theta}_S$

---

**Algorithm 2** Variance Reduced TD with Markovian samples Korda and La (2015)

**Input:** batch size $M$, learning rate $\alpha$ and initialization $\tilde{\theta}_0$
1: **for** $m = 1, 2, ..., S$ **do**
2:      $\theta_{m,0} = \tilde{\theta}_{m-1}$
3:      $g_m(\tilde{\theta}_{m-1}) = \frac{1}{M} \sum_{i=(m-1)M}^{mM-1} g_{x_i}(\tilde{\theta}_{m-1})$
4:      **for** $t = 0, 1, ..., M - 1$ **do**
5:          Sample $j_{m,t}$ uniformly at random in $\{(m-1)M, ..., mM - 1\}$ from trajetory
6:          $\theta_{m,t+1} = \Pi_{R_\theta} \Big(\theta_{m,t} + \alpha\big(g_{x_{j_{m,t}}}(\theta_{m,t})$
7:              $- g_{x_{j_{m,t}}}(\tilde{\theta}_{m-1}) + g_m(\tilde{\theta}_{m-1})\big)\Big)$
8:      **end for**
9:      set $\tilde{\theta}_m = \theta_{m,t}$ for randomly chosen $t \in \{1, 2, ..., M\}$
10: **end for**
**Output:** $\tilde{\theta}_S$

---

one sample from the batch, and updates the corresponding component in $g_m(\tilde{\theta}_{m-1})$. Here, $\Pi_{R_\theta}$ in Algorithm 2 denotes the projection operator onto a norm ball with the radius $R_\theta$. The idea is similar to the SVRG algorithm proposed in Johnson and Zhang (2013) for conventional optimization. Since a batch pseudo-gradient is used at each inner-loop update, the variance of the pseudo-gradient is expected to be reduced.

### 3.2 Technical Errors in Korda and La (2015)

In this subsection, we point out the technical errors in the analysis of VRTD in Korda and La (2015), which thus fails to provide the correct variance reduction performance for VRTD.

At the high level, the batch pseudo-gradient $g_m(\tilde{\theta}_{m-1})$ computed at the beginning of each epoch $m$ should necessarily introduce a non-vanishing variance error for a fixed stepsize, because it cannot exactly equal the mean (i.e. population) pseudo-gradient $g(\tilde{\theta}_{m-1})$. Furthermore, due to the correlation among samples, the pseudo-gradient estimator in expectation (with regard to the randomness of the sample trajectory) does not equal to the mean pseudo-gradient, which should further cause a non-vanishing bias error in the convergence bound. Unfortunately, the convergence bound in Korda and La (2015) indicates an exact convergence to the fixed point, which contradicts the aforementioned general understanding. More specifically, if the batch size $M = 1$ (with properly chosen $\lambda_A$ defined as $\lambda_A := 2|\lambda_{\max}(A + A^\top)|$), VRTD reduces to the vanilla TD. However, the exact convergence result in Theorem 3 in Korda and La (2015) does not agree with that of vanilla TD characterized in the recent studies Bhandari et al. (2018), which has variance and bias errors.

In Appendix B, we further provide a counter-example to show that one major technical step for characterizing the convergence bound in Korda and La (2015) does not hold. The goal of this paper is to provide a rigorous analysis of VRTD to characterize its variance reduction performance.

## 4 Main Results

As aforementioned, the convergence of VRTD consists of two types of errors: the variance error due to inexact estimation of the mean pseudo-gradient and the bias error due to Markovian sampling. In this section, we first focus on the first type of error and study the convergence of VRTD under i.i.d. sampling. We then study the Markovian case to further analyze the bias. In both cases, we compare the performance of VRTD to that of the vanilla TD described in eq. (1) to demonstrate its advantage.

### 4.1 CONVERGENCE ANALYSIS OF VRTD WITH I.I.D. SAMPLES

For i.i.d. samples, it is expected that the bias error due to the time correlation among samples does not exist. However, if we directly apply VRTD (Algorithm 2) originally designed for Markovian samples, there would be a bias term due to the correlation between the batch pseudo-gradient estimate and every inner-loop updates. Thus, we slightly modify Algorithm 2 to Algorithm 1 to avoid the bias error in the convergence analysis with i.i.d. samples. Namely, at each inner-loop iteration, we draw a new sample from the stationary distribution $\mu_\pi$ for the update rather than randomly selecting one from the batch of samples drawn at the beginning of the epoch as in Algorithm 2. In this way, the new independent samples avoid the correlation with the batch pseudo-gradient evaluated at the beginning of the epoch. Hence, Algorithm 1 does not suffer from an extra bias error.

To understand the convergence of Algorithm 1 at the high level, we first note that the sample batch pseudo-gradient cannot estimate the mean pseudo-gradient $g(\tilde{\theta}_{m-1})$ exactly due to its population nature. Then, we define $e_m(\tilde{\theta}_m) = g_m(\tilde{\theta}_{m-1}) - g(\tilde{\theta}_{m-1})$ as such a pseudo-gradient estimation error, our analysis (see Appendix D) shows that after each epoch update, we have

$$\mathbb{E}\left[\left\|\tilde{\theta}_m - \theta^*\right\|_2^2 \Big| F_{m,0}\right]$$
$$\leq \frac{1/M + 4\alpha^2(1+\gamma)^2}{\alpha\lambda_A - 4\alpha^2(1+\gamma)^2}\left\|\tilde{\theta}_{m-1} - \theta^*\right\|_2^2 + \frac{2\alpha}{\lambda_A - 4\alpha(1+\gamma)^2}\mathbb{E}\left[\left\|e_m(\tilde{\theta}_{m-1})\right\|_2^2 \Big| F_{m,0}\right], \quad (2)$$

where $F_{m,0}$ denotes the $\sigma$-field that includes all the randomness in sampling and updates before the $m$-th epoch. The first term in the right-hand side of eq. (2) captures the contraction property of Algorithm 1 and the second term corresponds to the variance of the pseudo-gradient estimation error. It can be seen that due to such an error term, Algorithm 1 is expected to have guaranteed convergence only to a neighborhood of $\theta^*$, when applying eq. (2) iteratively. Our further analysis shows that such an error term can still be well controlled (to be small) by choosing an appropriate value for the batch size $M$, which captures the advantage of the variance reduction. The following theorem precisely characterizes the non-asymptotic convergence of Algorithm 1.

**Theorem 1.** *Consider the VRTD algorithm in Algorithm 1. Suppose Assumptions 1–3 hold. Set a constant stepsize $\alpha < \frac{\lambda_A}{8(1+\gamma)^2}$ and the batch size $M > \frac{4(1+\gamma)^2\alpha^2+1}{\alpha[\lambda_A-8\alpha(1+\gamma)^2]}$. Then, for all $m \in \mathbb{N}$,*

$$\mathbb{E}\left[\left\|\tilde{\theta}_m - \theta^*\right\|_2^2\right] \leq C_1^m \left\|\tilde{\theta}_0 - \theta^*\right\|_2^2 + \frac{2D_2\alpha}{(1-C_1)(\lambda_A - 4\alpha(1+\gamma)^2)M}, \quad (3)$$

*where $C_1 := \left(4\alpha(1+\gamma)^2 + \frac{4(1+\gamma)^2\alpha^2+1}{\alpha M}\right)\frac{1}{\lambda_A-4\alpha(1+\gamma)^2}$ (with $C_1 < 1$ due to the choices of $\alpha$ and $M$), and $D_2 = 4((1+\gamma)^2R_\theta^2 + r_{\max}^2)$.*

We note that the convergence rate in eq. (3) can be written in a simpler form as $\mathbb{E}[||\tilde{\theta}_m - \theta^*||^2] \leq C_1^m||\tilde{\theta}_0 - \theta^*||^2 + \mathcal{O}(\alpha/M)$.

Theorem 1 shows that Algorithm 1 converges linearly (under a properly chosen constant stepsize) to a neighborhood of the fixed point solution, and the size of the neighborhood (i.e., the error term) has the order of $\mathcal{O}(\frac{\alpha}{M})$, which can be made as small as possible by properly increasing the batch size $M$. This is in contrast to the convergence result of the vanilla TD, which suffers from the constant error term with order $\mathcal{O}(\alpha)$ Bhandari et al. (2018) for a fixed stepsize. Thus, a small stepsize $\alpha$ is required in vanilla TD to reduce the variance error, which, however, slows down the practical convergence significantly. In contrast, this is not a problem for VRTD, which can attain a high accuracy solution while still maintaining fast convergence at a desirable stepsize.

We further note that if we have access to the mean pseudo-gradient $g(\tilde{\theta}_{m-1})$ in each epoch $m$, then the error term $||\theta_m - \theta^*||_2^2$ becomes zero, and Algorithm 1 converges linearly to the exact fixed point solution, as the iteration number $m$ goes to infinity with respect to the conditional number $C_1$, which is a positive constant and less than 1. This is similar to the conventional convergence of SVRG for strongly convex optimization Johnson and Zhang (2013). However, the proof here is very different. In Johnson and Zhang (2013), the convergence proof relies on the relationship between the gradient and the value of the objective function, but there is not such an objective function in the TD learning problem. Thus, the convergence of the parameter $\theta$ needs to be developed by exploiting the structure of the Bellman operator.

Based on the convergence rate of VRTD characterized in Theorem 1 under i.i.d. sampling, we obtain the following bound on the overall computational complexity.

**Corollary 1.** *Suppose Assumptions 1-3 hold. Let $\alpha = \frac{\lambda_A}{16(1+\gamma)^2}$, $M = \lceil \max\{\frac{D_2}{3(1-C_1)} \frac{1}{\epsilon}, \frac{33(1+\gamma)^2}{\lambda_A^2}\} \rceil$. Then, for any $\epsilon > 0$, an $\epsilon$-accuracy solution (i.e., $\mathbb{E}||\tilde{\theta}_m - \theta^*||^2 \leq \epsilon$) can be attained with at most $m = \lceil \log \frac{2\|\tilde{\theta}_0 - \theta^*\|_2^2}{\epsilon} / \log \frac{1}{C_1} \rceil$ iterations. Correspondingly, the total number of pseudo-gradient computations required by VRTD (i.e., Algorithm 1) under i.i.d. sampling to attain such an $\epsilon$-accuracy solution is at most*

$$\mathcal{O}\left( \max\left\{ \frac{1}{\epsilon}, \frac{1}{\lambda_A^2} \right\} \log \left( \frac{1}{\epsilon} \right) \right).$$

*Proof.* Given the values of $\alpha$ and $M$ in the theorem, it can be easily checked that $\mathbb{E}||\tilde{\theta}_m - \theta^*||^2 \leq \epsilon$ for $m = \lceil \log \frac{2\|\tilde{\theta}_0 - \theta^*\|_2^2}{\epsilon} / \log \frac{1}{C_1} \rceil$. Then the total number of pseudo-gradient computations is given by $2mM$ that yields the desired order given in the theorem. $\square$

As a comparison, consider the vanilla TD algorithm studied in Bhandari et al. (2018) with the constant stepsize $\alpha = O(\epsilon)$. If the samples are i.i.d. generated, it can be shown (see Appendix F.1) that the vanilla TD requires $\mathcal{O}\left( \frac{1}{\epsilon\lambda_A^2} \log \left( \frac{1}{\epsilon} \right) \right)$ pseudo-gradient computations in total to obtain an $\epsilon$-accuracy solution. Clearly, VRTD has lower computational complexity than vanilla TD if $\lambda_A$ is small. Such a comparison is similar in nature to the comparison between SVRG Johnson and Zhang (2013) and SGD in traditional optimization, where SVRG achieves better computational complexity than SGD for strongly convex objectives if the conditional number of the loss is small.

## 4.2 CONVERGENCE ANALYSIS OF VRTD WITH MAKOVIAN SAMPLES

In this section, we study the VRTD algorithm (i.e., Algorithm 2) with Markovian samples, in which samples are generated from one single MDP path. In such a case, we expect that the convergence of VRTD to have both the variance error due to the pseudo-gradient estimation (similar to the case with i.i.d. samples) and the bias error due to the correlation among samples. To understand this at the high level, we define the bias at each iteration as $\xi_m(\theta) = (\theta - \theta^*)^\top (g_m(\theta) - g(\theta))$. Then our analysis (see Appendix E) shows that after the update of each epoch, we have

$$
\begin{aligned}
&\mathbb{E}\left[ \left\| \tilde{\theta}_m - \theta^* \right\|_2^2 \Big| F_{m,0} \right] \\
&\leq \frac{1/M + 3\alpha^2(1+\gamma)^2}{\alpha\lambda_A - 3\alpha^2(1+\gamma)^2} \left\| \tilde{\theta}_{m-1} - \theta^* \right\|_2^2 + \frac{3\alpha}{\lambda_A - 3\alpha(1+\gamma)^2} \mathbb{E}\left[ \|g_m(\theta^*)\|_2^2 \Big| F_{m,0} \right] \\
&\quad + \frac{2}{[\lambda_A - 3\alpha(1+\gamma)^2]M} \sum_{i=0}^{M-1} \mathbb{E}\left[ \xi_m(\theta_{m,i}) \Big| F_{m,0} \right]
\end{aligned}
\tag{4}
$$

The first term on the right-hand side of eq. (4) captures the epochwise contraction property of Algorithm 2. The second term is due to the variance of the pseudo-gradient estimation, which captures how well the batch pseudo-gradient $g_m(\theta^*)$ approximates the mean pseudo-gradient $g(\theta^*)$ (note that $g(\theta^*) = 0$). Such a variance term can be shown to decay to zero as the batch size gets large similarly to the i.i.d. case. The third term captures the bias introduced by the correlation among samples in the $m$-th epoch. To quantitatively understand this error term, we provide the following lemma that characterizes how the bias error is controlled by the batch size $M$.

**Lemma 1.** *For any $m > 0$ and any $\theta \in B_\theta$, which is a ball with the radius $R_\theta$, we have*

$$\mathbb{E}[\xi_m(\theta)] \leq \frac{\lambda_A}{4} \mathbb{E}[\|\theta - \theta^*\|_2^2 | \mathcal{F}_{n,0}] + \frac{8[1 + (\kappa - 1)\rho]}{\lambda_A(1-\rho)M} [R_\theta^2(1+\gamma)^2 + r_{\max}^2],$$

*where the expectation is over the random trajectory, $\theta$ is treated as a fixed variable, and $0 < C_0 < \infty$ is a constant depending only on the MDP.*

Lemma 1 shows that the bias error diminishes as the batch size $M$ increases and the algorithm approaches to the fixed point $\theta^*$. To explain why this happens, the definition of $\xi_m(\theta)$ immediately yields the following bound:

$$\xi_m(\theta) \le \frac{1}{\lambda_A} \|g_n(\theta) - g(\theta)\|_2^2 + \frac{\lambda_A}{4} \|\theta - \theta^*\|_2^2. \tag{5}$$

The first term on the right-hand-side of eq. (5) can be bounded by the concentration property for the ergodic process as $g_m(\theta) = \frac{1}{M} \sum_{i=(m-1)M}^{mM-1} g_{x_i}(\theta) \overset{a.s.}{\to} g(\theta)$. As $M$ increases, the randomness due to the pseudo-gradient estimation is essentially averaged out due to the variance reduction step in VRTD, which implicitly eliminates its correlation from samples in the previous epochs.

As a comparison, the bias error in vanilla TD has been shown to be bounded by $\mathbb{E}[\xi_n(\theta)] = \mathcal{O}(\alpha \log(1/\alpha))$ Bhandari et al. (2018). In order to reduce the bias and achieve a high convergence accuracy, the stepsize $\alpha$ is required to be small, which causes the algorithm to run very slowly. The advantage of VRTD is that the bias can be reduced by choosing a sufficiently large batch size $M$ so that the stepsize can still be kept at a desirable constant to guarantee fast convergence.

**Theorem 2.** *Consider the VRTD algorithm in Algorithm 2. Suppose Assumptions 1–3 hold. Set the constant stepsize $\alpha < \frac{\lambda_A}{12(1+\gamma)^2}$ and the batch size $M > \frac{1}{0.5\alpha\lambda_A - 6\alpha^2(1+\gamma)^2}$. Then, we have*

$$\mathbb{E}\left[\left\|\tilde{\theta}_m - \theta^*\right\|_2^2\right] \le C_1^m \left\|\tilde{\theta}_0 - \theta^*\right\|_2^2 + \frac{3C_4\alpha + C_2/\lambda_A}{(1-C_1)[0.5\lambda_A - 3\alpha(1+\gamma)^2]M}, \tag{6}$$

*where $C_1 = \frac{1/M + 3\alpha^2(1+\gamma)^2}{0.5\alpha\lambda_A - 3\alpha^2(1+\gamma)^2}$ (with $C_1 < 1$ due to the choices for $\alpha$ and $M$), $C_2 = \frac{16[1+(\kappa-1)\rho][R_\theta^2(1+\gamma)^2 + r_{\max}^2]}{1-\rho}$ and $C_4 = [(1+\gamma)R_\theta + r_{\max}]^2 + \frac{2\rho\kappa G[(1+\gamma)R_\theta + r_{\max}]}{1-\rho}$.*

We note that the convergence rate in eq. (6) can be written in a simpler form as $\mathbb{E}[\|\tilde{\theta}_m - \theta^*\|^2] \le C_1^m \|\tilde{\theta}_0 - \theta^*\|^2 + \mathcal{O}(1/M)$.

Theorem 2 shows that VRTD (i.e., Algorithm 2) with Markovian samples converges to a neighborhood of $\theta^*$ at a linear rate, and the size of the neighborhood (i.e., the convergence error) decays sublinearly with the batch size $M$. More specifically, the first term in the right-hand side of eq. (6) captures the linear convergence of the algorithm, the second term corresponds to the sum of the cumulative pseudo-gradient estimation error and the cumulative bias error. For the fixed stepsize, the total convergence error is dominated by the sum of those two error terms with the order $\mathcal{O}(1/M)$. Therefore, the variance reduction in Algorithm 2 reduces both the variance and the bias of the pseudo-gradient estimator.

Based on the convergence rate of VRTD characterized in Theorem 2 under Markovian sampling, we obtain the following bound on the corresponding computational complexity.

**Corollary 2.** *Suppose Assumptions 1-3 hold. Let $\alpha = \frac{\lambda_A}{24(1+\gamma)^2}$ and $M = \lceil (\frac{32C_2 + 4C_4}{3(1-C_1)} + 100(1+\gamma)^2) \max\{\frac{1}{\epsilon}, \frac{1}{\epsilon\lambda_A^2}\} \rceil$. Then, for any $\epsilon > 0$, an $\epsilon$-accuracy solution (i.e., $\mathbb{E}\|\tilde{\theta}_m - \theta^*\|^2 \le \epsilon$) can be attained with at most $m = \lceil \log \frac{2\|\tilde{\theta}_0 - \theta^*\|_2^2}{\epsilon} / \log \frac{1}{C_1} \rceil$ iterations. Correspondingly, the total number of pseudo-gradient computations required by VRTD (i.e., Algorithm 2) under Markovian sampling to attain such an $\epsilon$-accuracy solution is at most*

$$\mathcal{O}\left(\max\left\{\frac{1}{\epsilon}, \frac{1}{\epsilon\lambda_A^2}\right\} \log \frac{1}{\epsilon}\right).$$

*Proof.* Given the values of $\alpha$ and $M$ in the theorem, it can be easily checked that $\mathbb{E}\|\tilde{\theta}_m - \theta^*\|^2 \le \epsilon$ for $m = \lceil \log \frac{2\|\tilde{\theta}_0 - \theta^*\|_2^2}{\epsilon} / \log \frac{1}{C_1} \rceil$. Then the total number of pseudo-gradient computations is given by $2mM$ that yields the desired order given in the theorem. $\square$

As a comparison, consider the vanilla TD algorithm studied in Bhandari et al. (2018) with the constant stepsize $\alpha = O(\epsilon/\log(1/\epsilon))$. Under Markovian sampling, it can be shown (see Appendix F.2) that vanilla TD requires $\mathcal{O}\left(\frac{1}{\epsilon\lambda_A^2} \log^2\left(\frac{1}{\epsilon}\right)\right)$ pseudo-gradient computations in total to obtain an $\epsilon$-accuracy

solution. Hence, in the Markovian setting, VRTD outperforms vanilla TD in terms of the total computational complexity by a factor of $\log \frac{1}{\epsilon}$. To intuitively explain, we first note that the correlation among data samples in the Markovian case also causes a bias error in addition to the variance error. For VRTD, due to the variance reduction scheme, the bias and variance errors are kept at the same level (with respect to the batch size) so that the bias error does not cause order-level increase in the computational complexity for VRTD. However, for vanilla TD, the bias error dominates the variance error, which turns out to require more iterations to attain an $\epsilon$-accurate solution, and yields an additional $\log \frac{1}{\epsilon}$ factor in the total complexity compared to VRTD.

## 5 EXPERIMENTS

In this section, we provide numerical results to verify our theoretical results. Note that in Appendix A, we provide further experiments on two problems in OpenAI Gym Brockman et al. (2016) and one experiment to demonstrate that VRTD is more sample-efficient than vanilla TD.

We consider an MDP with $\gamma = 0.95$ and $|S| = 50$. Each transition probability are randomly sampled from [0,1] and the transitions were normalized to one. The expected reward for each transition is also generated randomly in [0,1] and the reward on each transition was sampled without noise. Each component of the feature matrix $\Phi \in \mathbb{R}^{50 \times 4}$ is randomly and uniformly sampled between 0 and 1. The baseline for comparison is the vanilla TD algorithm, which corresponds to the case with $M = 1$ in our figure. We conduct two experiments to investigate how the batch size $M$ for variance reduction affects the performance of VRTD with i.i.d. and Markovian samples. In the Markovian setting, we sample the data from a MDP trajectory. In the i.i.d. setting, we sample the data independently from the corresponding stationary distribution. In both experiments, we set the constant stepsize to be $\alpha = 0.1$ and we run the experiments for five different batch sizes: $M = 1, 50, 500, 1000, 2000$. Our results are reported in Figure 1 and 2. All the plots report the square error over 1000 independent runs. In each case, the left figure illustrates the convergence process over the number of pseudo-gradient computations and the right figure shows the convergence errors averaged over the last 10000 iterations for different batch size values. It can be seen that in both i.i.d. and Markovian settings, the averaged error decreases as the batch size increases, which corroborates both Theorem 1 and Theorem 2. We also observe that increased batch size substantially reduces the error without much slowing down the convergence, demonstrating the desired advantage of variance reduction. Moreover, we observe that the error of VRTD with i.i.d samples is smaller than that of VRTD with Markovian samples under all batch size settings, which indicates that the correlation among Markovian samples introduces additional errors.

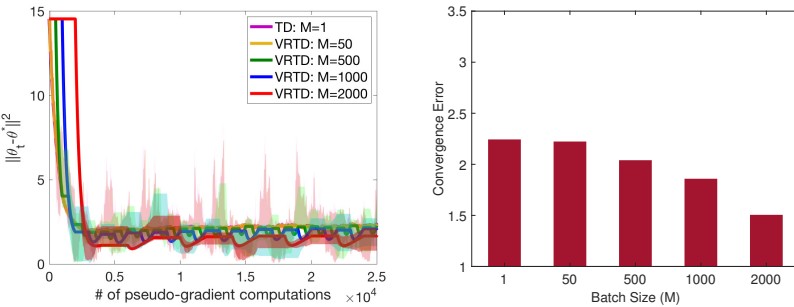

(a) left: iteration process; right: averaged convergence error

Figure 1: Error decay of VRTD with i.i.d. sample

## 6 CONCLUSION

In this paper, we provided the convergence analysis for VRTD with both i.i.d. and Markovian samples. We developed a novel technique to bound the bias of the VRTD pseudo-gradient estimator. Our result demonstrate the advantage of VRTD over vanilla TD on the reduced variance and bias errors by the batch size. We anticipate that such a variance reduction technique and our analysis tools can be further applied to other RL algorithms.

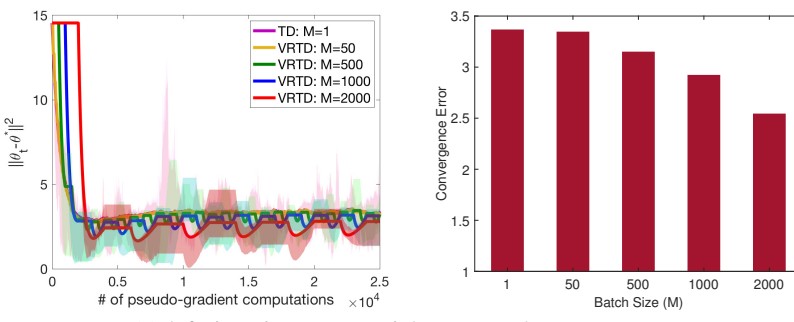

(a) left: iteration process; right: averaged convergence error

Figure 2: Error decay of VRTD with Markovian sample

## ACKNOWLEDGMENTS

The work was supported in part by US National Science Foundation under the grants CCF-1801855, ECCS-1818904, and CCF-1909291. The authors would like to thank Bowen Weng at the Ohio State University for the helpful discussions on the experiments. The authors would also like to thank a few anonymous reviewers for their suggestions on the analysis of the overall computational complexity as well as additional experiments, which significantly help to improve the quality of the paper.

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

# Supplementary Materials

## A    ADDITIONAL EXPERIMENTS

In this section, we assess the practical performance of VRTD (Algorithm 2) on two problems in OpenAI Gym Brockman et al. (2016), which are Frozen Lake ($4 \times 4$) and Mountain Car. We also provide an additional experiment to demonstrate that VRTD is more sample-efficient than vanilla TD.

### A.1    FROZEN LAKE

Frozen Lake is a game in OpenAI Gym, which is designed by an MDP with a finite state and action space. An agent starts from the starting point at $t = 0$ and can only transport to the neighbor blocks. It returns to the start-point every time it reaches a "hole" or the "goal". The agent receives a reward 1 only when it reaches the goal and 0 otherwise. Each transition probability is randomly sampled from $[0, 1]$ and normalized to one, and each component of the feature matrix $\Phi \in \mathbb{R}^{16 \times 4}$ is also randomly sampled from $[0, 1]$. Given the feature matrix and the transition probability, the ground truth value of $\theta^*$ can be calculated, which is used to evaluate the error in the experiments. We set the stepsize to be $\alpha = 0.1$ and run vanilla TD ($M = 1$) and VRTD with the batch sizes $M = 50, 500, 1000, 2000$. Note that $M = 1$ corresponds to the base line vanilla TD. We compute the squared error over 1000 independent runs. The left plot in Figure 3 shows the convergence process over the number of pseudo-gradient computations and the right plot in Figure 3 shows the convergence error averaged over the last 10000 iterations. It can be observed that VRTD achieves much smaller error than TD, and increasing the batch size for VRTD substantially reduces the error without much slowing down the convergence.

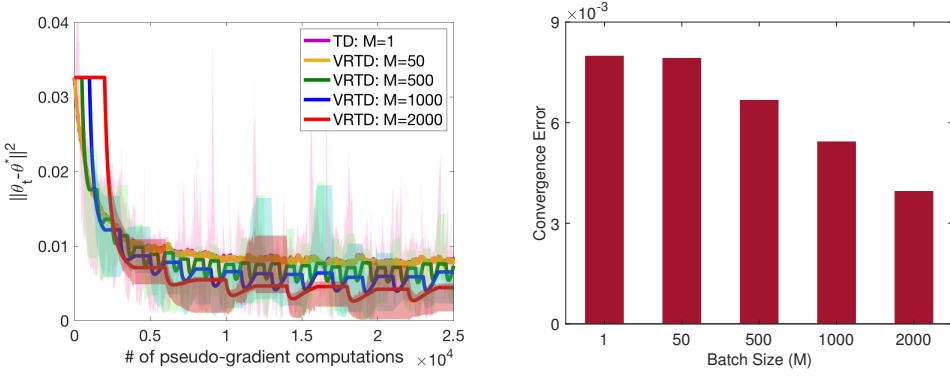

(a) left: iteration process; right: averaged convergence error

Figure 3: Error decay of VRTD in Frozen Lake problem

### A.2    MOUNTAIN CAR

Mountain Car is a game in OpenAI Gym, which is driven by an MDP with an infinite state space and a finite action space. At each time step, an agent randomly chooses an action $\in \{\text{push left}, \text{push right}, \text{no push}\}$. In this problem, the ground truth value of $\theta^*$ is not known. In order to quantify the performance of VRTD, we apply the error metric known as the norm of the expected TD update given by NEU$= \|\mathbb{E}[\delta\phi]\|_2^2$, where $\delta$ is the temporal difference Sutton et al. (2009); Maei (2011). The state sample is transformed into a feature vector with the dimension 20 using an approximation of a RBF kernel. The agent follows a random policy in our experiment and we initialize $\theta_0 = 0$. At $t = 0$, the agent starts from the lowest point, receives a reward of $-1$ at each time step, and returns to the starting point every time it reaches the goal. We set the stepsize to be $\alpha = 0.2$ and run vanilla TD ($M = 1$) and VRTD with batch size $M = 1000$. After every 10000 pseudo-gradient computations, learning is paused and the NEU is computed by averaging over 1000 test samples. We conduct 1000 independent runs and the results are reported by averaging over these

runs. Figure 4 shows the convergence process of the NEU versus the number of pseudo-gradient computations. It can been seen that VRTD achieves smaller NEU than vanilla TD.

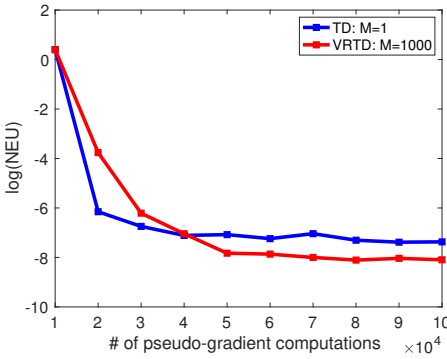

Figure 4: NEU decay of VRTD in Mountain Car problem

### A.3 COMPARISON BETWEEN VRTD AND TD WITH A CHANGING STEPSIZE

In this subsection, we provide an additional experiment to compare the performance of VRTD given in Algorithm 2 (under constant stepsize) with the TD algorithm (under a changing stepsize as suggested by the reviewer). We adopt the same setting of Frozen Lake as in Appendix A.1. Let VRTD take a batch size $M = 5000$ and stepsize $\alpha = 0.1$. For a fair comparison, we start TD with the same constant stepsize $\alpha = 0.1$ and then reduce the stepsize by half whenever the error stops decrease. The comparison is reported in Figure 5, where both curves are averaged over 1000 independent runs. The two algorithms are compared in terms of the squared error versus the total number of pseudo-gradient computations (equivalently, the total number of samples being used). It can be seen that VRTD reaches the required accuracy much faster than TD.

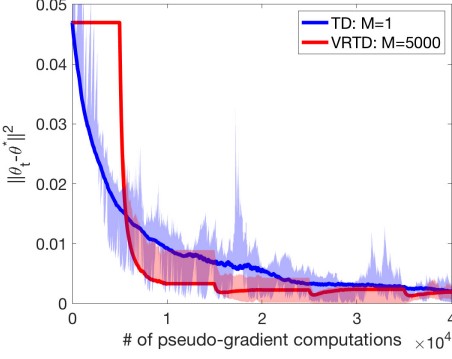

Figure 5: Comparison of error decay of VRTD and TD with changing stepsize in Frozen Lake problem

## B A COUNTER EXAMPLE

In this section, we use a counter-example to show that one major technical step for characterizing the convergence bound in Korda and La (2015) does not hold. Consider Step 4 in the proof of Theorem 3 in Korda and La (2015). For the following defined $\epsilon(\theta)$

$$\epsilon(\theta) = (\theta - \theta^*)^\top [\mathbb{E}(v^\top v | \mathcal{F}_n) - \mathbb{E}_{\Psi, \theta_n}(v^\top v)](\theta - \theta^*), \tag{7}$$

where $\Psi$ denotes the stationary distribution of the corresponding Markov chain, Korda and La (2015) claimed that the following inequality holds

$$\|\epsilon(\theta)\|_2 \le 2H \|\mathbb{E}(v|\mathcal{F}_n) - \mathbb{E}_{\Psi,\theta_n}(v)\|_2. \tag{8}$$

This is not correct. Consider the following counter-example. Let the batch size $M = 3$ and the dimension of the feature vector be one, i.e., $\Phi \in \mathbb{R}^{|\mathcal{S}| \times 1}$. Hence, all variables in eq. (8) and eq. (7) are scalars. Since the steps for proving eq. (8) in Korda and La (2015) do not have specific requirements for the transition kernel, eq. (8) should hold for any distribution of $v$. Thus, suppose $v$ follows the uniform distribution over $[-3, 3]$. Further assume that in the $n$-th epoch, the samples of $v$ are given by $\{1, 2, -3\}$. Recall that $\mathbb{E}(\cdot|\mathcal{F}_n)$ is the average over the batch samples in the $n$-th epoch. We have:

$$\mathbb{E}_{\Psi,\theta_n}(v) = 0, \quad \mathbb{E}_{\Psi,\theta_n}(v^2) = 3, \quad \mathbb{E}(v|\mathcal{F}_n) = 0, \quad \mathbb{E}(v^2|\mathcal{F}_n) = \frac{14}{3}.$$

Substituting the above values into eq. (8) yields

$$\|\epsilon(\theta)\|_2 = \left(\frac{14}{3} - 3\right)(\theta - \theta^*)^2 \le 2H \times 0 = 0, \tag{9}$$

which obviously does not hold in general when $\theta \ne \theta^*$. Consequently the second statement in Theorem 3 of Korda and La (2015), which is critically based on the above erroneous steps, does not hold. Hence, the first statement in the same theorem whose proof is based on the second statement cannot hold either.

## C  USEFUL LEMMAS

In the rest of the paper, for any matrix $W \in \mathbb{R}^{d \times d}$, we denote $\|W\|_2$ as the spectral norm of $W$ and $\|W\|_F$ as the Frobenius norm of $W$.

**Lemma 2.** *For any $x_i = (s_i, r_i, s_i')$ (i.i.d. sample) or $x_i = (s_i, r_i, s_{i+1})$ (Markovian sample), we have $\|A_{x_i}\|_2 \le 1 + \gamma$ and $\|b_{x_i}\|_2 \le r_{\max}$.*

*Proof.* First consider the case when samples are i.i.d. Due to the definition of $A_{x_i}$, we have

$$\begin{aligned}
\|A_{x_i}\|_2 &= \left\|\phi(s_i)(\gamma\phi(s_i') - \phi(s_i))^\top\right\|_2 \\
&\le \left\|\phi(s_i)(\gamma\phi(s_i') - \phi(s_i))^\top\right\|_F \\
&\le \gamma \left\|\phi(s_i)\phi(s_i')^\top\right\|_F + \left\|\phi(s_i)\phi(s_i)^\top\right\|_F \\
&\le 1 + \gamma.
\end{aligned}$$

Then, consider $b_{x_i}$:

$$\|b_{x_i}\|_2 = \|r_{x_i}\phi(s_i)\|_2 \le r_{\max}\|\phi(s_i)\|_2 \le r_{\max}.$$

Following similar steps, we can obtain the same upper bounds for the case with Markovian samples. $\square$

**Lemma 3.** *Let $G = (1 + \gamma)R_\theta + r_{\max}$. Consider Algorithm 2. For any $m > 0$ and $0 \le t \le M - 1$, we have $\left\|g_{x_{j_m,t}}(\theta_{m,t})\right\|_2, \left\|g_{x_{j_m,t}}(\tilde{\theta}_{m-1})\right\|_2, \left\|g_m(\tilde{\theta}_{m-1})\right\|_2 \le G$.*

*Proof.* First, we bound $\left\|g_{x_{j_m,t}}(\theta_{m,t})\right\|_2$ as follows.

$$\begin{aligned}
\left\|g_{x_{j_m,t}}(\theta_{m,t})\right\|_2 &= \left\|A_{x_{j_m,t}}\theta_{m,t} + b_{\theta_{m,t}}\right\|_2 \\
&\le \left\|A_{x_{j_m,t}}\right\|_2 \|\theta_{m,t}\|_2 + \left\|b_{\theta_{m,t}}\right\|_2 \\
&\le (1 + \gamma)R_\theta + r_{\max}.
\end{aligned}$$

Following the steps similar to the above, we have $\left\|g_{x_{j_m,t}}(\tilde{\theta}_{m-1})\right\|_2 \le G$. Finally for $\left\|g_{x_{j_m,t}}(\tilde{\theta}_{m-1})\right\|_2$, we have

$$\left\|g_{x_{j_m,t}}(\tilde{\theta}_{m-1})\right\|_2 = \left\|\frac{1}{M}\sum_{i=(m-1)M}^{mM-1} g_{x_i}(\tilde{\theta}_{m-1})\right\|_2$$

$$\leq \frac{1}{M} \sum_{i=(m-1)M}^{mM-1} \left\| g_{x_i}(\tilde{\theta}_{m-1}) \right\|_2$$

$$\leq G, \tag{10}$$

where eq. (10) follows from the last fact $\left\| g_{x_{j_m,t}}(\tilde{\theta}_{m-1}) \right\|_2 \leq G$. $\qquad \square$

**Lemma 4.** *Define* $D_1 = 2(1+\gamma)^2$ *and* $D_2 = 4((1+\gamma)^2 R_\theta^2 + r_{\max}^2)$. *For any* $\theta \in \mathbb{R}^d$, *we have* $\|g_{x_i}(\theta)\|_2^2 \leq D_1 \|\theta - \theta^*\|_2^2 + D_2$.

*Proof.* Recalling the definition of $g_{x_i}$, and applying Lemma 2, we have

$$
\begin{aligned}
\|g_{x_i}(\theta)\|_2^2 &= \|A_{x_i}\theta + b_{x_i}\|_2^2 \\
&= \|A_{x_i}(\theta - \theta^*) + (A_{x_i}\theta^* + b_{x_i})\|_2^2 \\
&\leq 2\|A_{x_i}(\theta - \theta^*)\|_2^2 + 2\|A_{x_i}\theta^* + b_{x_i}\|_2^2 \\
&\leq 2\|A_{x_i}\|_2^2 \|\theta - \theta^*\|_2^2 + 4(\|A_{x_i}\|_2^2 \|\theta^*\|_2^2 + \|b_{x_i}\|_2^2) \\
&\leq 2(1+\gamma)^2 \|\theta - \theta^*\|_2^2 + 4((1+\gamma)^2 R_\theta^2 + r_{\max}^2) \\
&= D_1 \|\theta - \theta^*\|_2^2 + D_2.
\end{aligned}
$$

$\qquad \square$

**Lemma 5.** *Considering Algorithm 2 with Markovian samples. We have* $\|\mathbb{E}[A_j|P_i] - A\|_F \leq (1+\gamma)\kappa\rho^{j-i}$ *and* $\|\mathbb{E}[b_j|P_i] - b\|_2 \leq r_{\max}\kappa\rho^{j-i}$ *for* $0 < i < j$.

*Proof.* We first derive

$$
\begin{aligned}
\|\mathbb{E}[A_j|P_i] - A\|_F &= \left\| \int A_{x_i} dP(x_i|P_j) - \int A_{x_i} d\mu_\pi \right\|_F \\
&\leq \int \|A_{x_i} dP(x_i|P_j) - A_{x_i} d\mu_\pi\|_F \\
&\leq \int \|A_{x_i}\|_F |dP(x_i|P_j) - d\mu_\pi| \\
&\leq (1+\gamma) \|P(x_i|P_j), \mu_\pi\|_{TV} \\
&\leq (1+\gamma)\kappa\rho^{j-i}.
\end{aligned}
$$

Following the steps similar to the above, we can derive $\|\mathbb{E}[b_j|P_i] - b\|_2 \leq 2r_{\max}\kappa\rho^{j-i}$. $\qquad \square$

## D  PROOF OF THEOREM 1: CONVERGENCE OF VRTD WITH I.I.D. SAMPLES

Recall that $B_m$ is the sample batch drawn at the beginning of each $m$-th epoch and $x_{i,j}$ denotes the sample picked at the $j$-th iteration in the $i$-th epoch in Algorithm 1. We denote $\sigma(\tilde{\theta}_0)$ as a trivial $\sigma$-field when $\tilde{\theta}_0$ is a deterministic vector. Let $\sigma(A \cup B)$ indicate the smallest $\sigma$-field that contains both $A$ and $B$. Then, we construct a set of $\sigma$-fields in the following incremental way.

$F_{1,0} = \sigma(\tilde{\theta}_0)$, $F_{1,1} = \sigma(F_{1,0} \cup \sigma(B_1) \cup \sigma(x_{1,1}))$, ..., $F_{1,M} = \sigma(F_{1,(M-1)} \cup \sigma(x_{1,M}))$,

$F_{2,0} = \sigma(F_{1,M} \cup \sigma(\tilde{\theta}_1))$, $F_{21} = \sigma(F_{2,0} \cup \sigma(B_2) \cup \sigma(x_{2,1}))$, ..., $F_{2,m} = \sigma(F_{2,(M-1)} \cup \sigma(x_{2,M}))$,

$\vdots$

$F_{m,0} = \sigma(F_{(m-1),M} \cup \sigma(\tilde{\theta}_{m-1}))$, $F_{m1} = \sigma(F_{m,0} \cup \sigma(B_m) \cup \sigma(x_{m,1}))$, ..., $F_{m,M} = \sigma(F_{m,(M-1)} \cup \sigma(x_{m,M}))$.

The proof of Theorem 1 proceeds along the following steps.

**Step 1: Iteration within the $m$-th epoch**

For the $m$-th epoch, we consider the last update (i.e., the $M$-th iteration in the epoch), and decompose its error into the following form.

$$\|\theta_{m,M} - \theta^*\|_2^2 = \left\|\theta_{m,M-1} + \alpha\left(g_{x_{j_m,M}}(\theta_{m,M-1}) - g_{x_{j_m,M}}(\tilde{\theta}_{m-1}) + g_m(\tilde{\theta}_{m-1})\right) - \theta^*\right\|_2^2$$

$$= \|\theta_{m,M-1} - \theta^*\|_2^2 + 2\alpha(\theta_{m,M-1} - \theta^*)^\top\left(g_{x_{j_m,M}}(\theta_{m,M-1}) - g_{x_{j_m,M}}(\tilde{\theta}_{m-1}) + g_m(\tilde{\theta}_{m-1})\right)$$

$$+ \alpha^2\left\|g_{x_{j_m,M}}(\theta_{m,M-1}) - g_{x_{j_m,M}}(\tilde{\theta}_{m-1}) + g_m(\tilde{\theta}_{m-1})\right\|_2^2. \tag{11}$$

First, consider the third term in the right-hand side of eq. (11), we have

$$\left\|g_{x_{j_m,M}}(\theta_{m,M-1}) - g_{x_{j_m,M}}(\tilde{\theta}_{m-1}) + g_m(\tilde{\theta}_{m-1})\right\|_2^2$$

$$\leq 2\left\|g_{x_{j_m,M}}(\theta_{m,M-1}) - g_{x_{j_m,M}}(\tilde{\theta}_{m-1}) + g(\tilde{\theta}_{m-1})\right\|_2^2 + 2\left\|g_m(\tilde{\theta}_{m-1}) - g(\tilde{\theta}_{m-1})\right\|_2^2$$

$$= 2\left\|g_{x_{j_m,M}}(\theta_{m,M-1}) - g_{x_{j_m,M}}(\theta^*) - \left[(g_{x_{j_m,M}}(\tilde{\theta}_{m-1}) - g_{x_{j_m,M}}(\theta^*)) - (g(\tilde{\theta}_{m-1}) - g(\theta^*))\right]\right\|_2^2$$

$$+ 2\left\|g_m(\tilde{\theta}_{m-1}) - g(\tilde{\theta}_{m-1})\right\|_2^2$$

$$\leq 4\left\|g_{x_{j_m,M}}(\theta_{m,M-1}) - g_{x_{j_m,M}}(\theta^*)\right\|_2^2 + 4\left\|(g_{x_{j_m,M}}(\tilde{\theta}_{m-1}) - g_{x_{j_m,M}}(\theta^*)) - (g(\tilde{\theta}_{m-1}) - g(\theta^*))\right\|_2^2$$

$$+ 2\left\|g_m(\tilde{\theta}_{m-1}) - g(\tilde{\theta}_{m-1})\right\|_2^2. \tag{12}$$

Then, by taking the expectation conditioned on $F_{m,M-1}$ on both sides of eq. (12), we have

$$\mathbb{E}\left[\left\|g_{x_{j_m,M}}(\theta_{m,M-1}) - g_{x_{j_m,M}}(\tilde{\theta}_{m-1}) + g_m(\tilde{\theta}_{m-1})\right\|_2^2 \Big| F_{m,M-1}\right]$$

$$\overset{(i)}{\leq} 4\mathbb{E}\left[\left\|g_{x_{j_m,M}}(\theta_{m,M-1}) - g_{x_{j_m,M}}(\theta^*)\right\|_2^2 \Big| F_{m,M-1}\right]$$

$$+ 4\mathbb{E}\left[\left\|(g_{x_{j_m,M}}(\tilde{\theta}_{m-1}) - g_{x_{j_m,M}}(\theta^*)) - \mathbb{E}\left[g_{x_{j_m,M}}(\tilde{\theta}_{m-1}) - g_{x_{j_m,M}}(\theta^*)\big|F_{m,M-1}\right]\right\|_2^2 \Big| F_{m,M-1}\right]$$

$$+ 2\mathbb{E}\left[\left\|g_m(\tilde{\theta}_{m-1}) - g(\tilde{\theta}_{m-1})\right\|_2^2 \Big| F_{m,M-1}\right]$$

$$\overset{(ii)}{\leq} 4(1+\gamma)^2\mathbb{E}\left[\|\theta_{m,M-1} - \theta^*\|_2^2 |F_{m,M-1}\right] + 4(1+\gamma)^2\mathbb{E}\left[\left\|\tilde{\theta}_{m-1} - \theta^*\right\|_2^2 |F_{m,M-1}\right]$$

$$+ 2\mathbb{E}\left[\left\|g_m(\tilde{\theta}_{m-1}) - g(\tilde{\theta}_{m-1})\right\|_2^2 \Big| F_{m,M-1}\right]$$

where $(i)$ follows from the fact that $\mathbb{E}[(g_{x_{j_m,M}}(\tilde{\theta}_{m-1}) - g_{x_{j_m,M}}(\theta^*))|F_{m,M-1}] = g(\tilde{\theta}_{m-1}) - g(\theta^*)$, and $(ii)$ follows from the inequality $\mathbb{E}[(X - \mathbb{E}X)^2] \leq \mathbb{E}X^2$ and Lemma 2. Then, taking the expectation conditioned on $F_{m,M-1}$ on both sides of eq. (11) yields

$$\mathbb{E}\left[\|\theta_{m,M} - \theta^*\|_2^2 \Big| F_{m,M-1}\right]$$

$$= \|\theta_{m,M-1} - \theta^*\|_2^2 + 2\alpha(\theta_{m,M-1} - \theta^*)^\top\mathbb{E}\left[g_{x_{j_m,M}}(\theta_{m,M-1}) - g_{x_{j_m,M}}(\tilde{\theta}_{m-1}) + g_m(\tilde{\theta}_{m-1})\Big|F_{m,M-1}\right]$$

$$+ \alpha^2\mathbb{E}\left[\left\|g_{x_{j_m,M}}(\theta_{m,M-1}) - g_{x_{j_m,M}}(\tilde{\theta}_{m-1}) + g_m(\tilde{\theta}_{m-1})\right\|_2^2 \Big| F_{m,M-1}\right]$$

$$\overset{(i)}{\leq} \|\theta_{m,M-1} - \theta^*\|_2^2 + 2\alpha(\theta_{m,M-1} - \theta^*)^\top g(\theta_{m,M-1})$$

$$+ 2\alpha(\theta_{m,M-1} - \theta^*)^\top\left(\mathbb{E}\left[g_m(\tilde{\theta}_{m-1})\Big|F_{m,M-1}\right] - g(\tilde{\theta}_{m-1})\right)$$

$$+ 4\alpha^2(1+\gamma)^2\|\theta_{m,M-1} - \theta^*\|_2^2 + 4\alpha^2(1+\gamma)^2\left\|\tilde{\theta}_{m-1} - \theta^*\right\|_2^2$$

$$+ 2\alpha^2 \mathbb{E}\left[\left\|g_m(\tilde{\theta}_{m-1}) - g(\tilde{\theta}_{m-1})\right\|_2^2 \Big| F_{m,M-1}\right]$$

$$\overset{(ii)}{\leq} \|\theta_{m,M-1} - \theta^*\|_2^2 - \alpha\lambda_A \|\theta_{m,M-1} - \theta^*\|_2^2 + 2\alpha\mathbb{E}\left[\xi_m(\tilde{\theta}_{m-1}) \Big| F_{m,M-1}\right]$$

$$+ 4\alpha^2(1+\gamma)^2 \|\theta_{m,M-1} - \theta^*\|_2^2 + 4\alpha^2(1+\gamma)^2 \left\|\tilde{\theta}_{m-1} - \theta^*\right\|_2^2$$

$$+ 2\alpha^2 \mathbb{E}\left[\left\|g_m(\tilde{\theta}_{m-1}) - g(\tilde{\theta}_{m-1})\right\|_2^2 \Big| F_{m,M-1}\right]$$

$$\overset{(iii)}{\leq} \|\theta_{m,M-1} - \theta^*\|_2^2 - [\alpha\lambda_A - 4\alpha^2(1+\gamma)^2] \|\theta_{m,M-1} - \theta^*\|_2^2 + 4\alpha^2(1+\gamma)^2 \left\|\tilde{\theta}_{m-1} - \theta^*\right\|_2^2$$

$$+ 2\alpha\mathbb{E}\left[\xi_m(\tilde{\theta}_{m-1}) \Big| F_{m,M-1}\right] + 2\alpha^2 \mathbb{E}\left[\left\|g_m(\tilde{\theta}_{m-1}) - g(\tilde{\theta}_{m-1})\right\|_2^2 \Big| F_{m,M-1}\right], \tag{13}$$

where $(i)$ follows from the fact that $\mathbb{E}\left[g_{x_{j_{m,M}}}(\tilde{\theta}_{m-1}) \Big| F_{m,M-1}\right] = g(\tilde{\theta}_{m-1})$. In $(ii)$ we define $\lambda_A$ as the absolute value of the largest eigenvalue of matrix $(A^T + A)$, which is negative definite according to Tsitsiklis and Van Roy (1997). In $(iii)$ we define $\xi_m(\theta) = (\theta - \theta^*)^\top (g_m(\theta) - g(\theta))$ for $\theta \in \mathbb{R}^d$. Then, by applying eq. (13) iteratively, we have

$$\mathbb{E}\left[\|\theta_{m,1} - \theta^*\|_2^2 \Big| F_{m,0}\right]$$

$$\leq \|\theta_{m,0} - \theta^*\|_2^2 - [\alpha\lambda_A - 4\alpha^2(1+\gamma)^2] \sum_{i=0}^{M-1} \mathbb{E}\left[\|\theta_{m,i} - \theta^*\|_2^2 \Big| F_{m,0}\right] + 4M\alpha^2(1+\gamma)^2 \left\|\tilde{\theta}_{m-1} - \theta^*\right\|_2^2$$

$$+ 2\alpha M \mathbb{E}\left[\xi_m(\tilde{\theta}_{m-1}) \Big| F_{m,0}\right] + 2M\alpha^2 \mathbb{E}\left[\left\|g_m(\tilde{\theta}_{m-1}) - g(\tilde{\theta}_{m-1})\right\|_2^2 \Big| F_{m,0}\right]. \tag{14}$$

For all $1 \leq i \leq M$, we have

$$\mathbb{E}\left[\xi_m(\tilde{\theta}_{m-1}) \Big| F_{m,0}\right] = \mathbb{E}\left[g_m(\tilde{\theta}_{m-1}) \Big| F_{m,0}\right] - g(\tilde{\theta}_{m-1})$$

$$= \frac{1}{M} \sum_{i \in B_m} \mathbb{E}[A_{x_i}\tilde{\theta}_m + b_{x_i} | F_{m,0}] - (A\tilde{\theta}_m + b)$$

$$= \left[\left(\frac{1}{M} \sum_{i \in B_m} \mathbb{E}[A_{x_i}|F_{m,0}]\right) - A\right]\tilde{\theta}_m + \left[\left(\frac{1}{M} \sum_{i \in B_m} \mathbb{E}[b_{x_i}|F_{m,0}]\right) - b\right]$$

$$= 0.$$

Then, arranging terms in eq. (14) and using the above fact yield

$$[\alpha\lambda_A - 4\alpha^2(1+\gamma)^2] \sum_{i=0}^{M-1} \mathbb{E}\left[\|\theta_{m,i} - \theta^*\|_2^2 \Big| F_{m,0}\right]$$

$$\leq [1 + 4M\alpha^2(1+\gamma)^2] \left\|\tilde{\theta}_{m-1} - \theta^*\right\|_2^2 + 2M\alpha^2 \mathbb{E}\left[\left\|g_m(\tilde{\theta}_{m-1}) - g(\tilde{\theta}_{m-1})\right\|_2^2 \Big| F_{m,0}\right]. \tag{15}$$

Finally, dividing eq. (15) by $[\alpha\lambda_A - 4\alpha^2(1+\gamma)^2]M$ on both sides yields

$$\mathbb{E}\left[\left\|\tilde{\theta}_m - \theta^*\right\|_2^2 \Big| F_{m,0}\right]$$

$$\leq \frac{1/M + 4\alpha^2(1+\gamma)^2}{\alpha\lambda_A - 4\alpha^2(1+\gamma)^2} \left\|\tilde{\theta}_{m-1} - \theta^*\right\|_2^2 + \frac{2\alpha}{\lambda_A - 4\alpha(1+\gamma)^2} \mathbb{E}\left[\left\|g_m(\tilde{\theta}_{m-1}) - g(\tilde{\theta}_{m-1})\right\|_2^2 \Big| F_{m,0}\right]. \tag{16}$$

### Step 2: Bounding the variance error

For any $0 \leq k \leq m - 1$, we have

$$\mathbb{E}\left[\left\|g_m(\tilde{\theta}_{m-1}) - g(\tilde{\theta}_{m-1})\right\|_2^2 \Big| F_{m,0}\right] \tag{17}$$

$$
= \mathbb{E}\left[\left\|\frac{1}{M}\sum_{i\in B_m}g_{x_i}(\tilde{\theta}_{m-1}) - g(\tilde{\theta}_{m-1})\right\|_2^2\Big|F_{m,0}\right] = \frac{1}{M^2}\mathbb{E}\left[\left\|\sum_{i\in B_m}g_{x_i}(\tilde{\theta}_{m-1}) - g(\tilde{\theta}_{m-1})\right\|_2^2\Big|F_{m,0}\right]
$$

$$
= \frac{1}{M^2}\mathbb{E}\left[\Big(\sum_{i\in B_m}g_{x_i}(\tilde{\theta}_{m-1}) - g(\tilde{\theta}_{m-1})\Big)^\top\Big(\sum_{j\in B_m}g_{x_j}(\tilde{\theta}_{m-1}) - g(\tilde{\theta}_{m-1})\Big)\Big|F_{m,0}\right]
$$

$$
= \frac{1}{M^2}\sum_{i\in B_m}\sum_{j\in B_m}\mathbb{E}\left[\Big\langle g_{x_i}(\tilde{\theta}_{m-1}) - g(\tilde{\theta}_{m-1}), g_{x_j}(\tilde{\theta}_{m-1}) - g(\tilde{\theta}_{m-1})\Big\rangle\Big|F_{m,0}\right]
$$

$$
= \frac{1}{M^2}\sum_{i=j}\mathbb{E}\left[\left\|g_{x_i}(\tilde{\theta}_{m-1}) - g(\tilde{\theta}_{m-1})\right\|_2^2\Big|F_{m,0}\right]
$$

$$
= \frac{1}{M^2}\sum_{i=j}\mathbb{E}\left[\left\|g_{x_i}(\tilde{\theta}_{m-1}) - \mathbb{E}\big[g_{x_i}(\tilde{\theta}_{m-1})\big|F_{m,0}\big]\right\|_2\Big|F_{m,0}\right]
$$

$$
\leq \frac{1}{M^2}\sum_{i=j}\mathbb{E}\left[\left\|g_{x_i}(\tilde{\theta}_{m-1})\right\|_2^2\Big|F_{m,0}\right] \leq \frac{1}{M}\left(D_1\left\|\tilde{\theta}_{m-1} - \theta^*\right\|_2^2 + D_2\right), \tag{18}
$$

where eq. (18) follows from Lemma 4.

**Step 3: Iteration over $m$ epochs**

First, we substitute eq. (18) into eq. (16) to obtain

$$
\mathbb{E}\left[\left\|\tilde{\theta}_m - \theta^*\right\|_2^2\Big|F_{m,0}\right] \leq C_1\left\|\tilde{\theta}_{m-1} - \theta^*\right\|_2^2 + \frac{2D_2\alpha}{(\lambda_A - 4\alpha(1+\gamma)^2)M}, \tag{19}
$$

where we define $C_1 = \left(4\alpha(1+\gamma)^2 + \frac{2D_1\alpha^2+1}{\alpha M}\right)\frac{1}{\lambda_A - 4\alpha(1+\gamma)^2}$.

Taking the expectation of eq. (19) conditioned on $F_{m-1,0}$ and following the steps similar to those in step 1 to upper bound $\mathbb{E}\left[\left\|\tilde{\theta}_{m-1} - \theta^*\right\|_2^2\Big|F_{m-1,0}\right]$, we obtain

$$
\mathbb{E}\left[\left\|\tilde{\theta}_m - \theta^*\right\|_2^2\Big|F_{m-1,0}\right] \leq C_1\mathbb{E}\left[\left\|\tilde{\theta}_{m-1} - \theta^*\right\|_2^2\Big|F_{m-1,0}\right] + \frac{2D_2\alpha}{(\lambda_A - 4\alpha(1+\gamma)^2)M}
$$

$$
\leq C_1^2\left\|\tilde{\theta}_{m-2} - \theta^*\right\|_2^2 + \frac{2D_2\alpha}{(\lambda_A - 4\alpha(1+\gamma)^2)M}\sum_{k=0}^{1}C_1^k.
$$

Then, by following the above steps for $(m-1)$ times, we have

$$
\mathbb{E}\left[\left\|\tilde{\theta}_m - \theta^*\right\|_2^2\right] \leq C_1^m\left\|\tilde{\theta}_0 - \theta^*\right\|_2^2 + \frac{2D_2\alpha}{(\lambda_A - 4\alpha(1+\gamma)^2)M}\sum_{k=0}^{m-1}C_1^k
$$

$$
\leq C_1^m\left\|\tilde{\theta}_0 - \theta^*\right\|_2^2 + \frac{2D_2\alpha}{(1-C_1)(\lambda_A - 4\alpha(1+\gamma)^2)M},
$$

which yields the desirable result.

## E  PROOF OF THEOREM 2: CONVERGENCE OF VRTD WITH MARKOVIAN SAMPLES

We define $\sigma(S_i)$ to be the $\sigma$-field of all samples up to the $i$-th epoch and recall that $j_{m,t}$ is the index of the sample picked at the $t$-th iteration in the $m$-th epoch in Algorithm 2. Then we define a set of $\sigma$-fields in the following incremental way:

$F_{1,0} = \sigma(S_0)$, $F_{1,1} = \sigma(F_{1,0}\cup\sigma(j_{1,1}))$, ..., $F_{1,M} = \sigma(F_{1,(M-1)}\cup\sigma(j_{1,M}))$,

$F_{2,0} = \sigma(\sigma(S_1)\cup F_{1,M}\cup\sigma(\tilde{\theta}_1))$, $F_{21} = \sigma(F_{2,0}\cup\sigma(j_{2,1}))$, ..., $F_{2,m} = \sigma(F_{2,(M-1)}\cup\sigma(j_{2,M}))$,

$\vdots$

$F_{m,0} = \sigma(\sigma(S_{m-1})\cup F_{(m-1),M}\cup\sigma(\tilde{\theta}_{m-1}))$, $F_{m,1} = \sigma(F_{m,0}\cup\sigma(j_{m,1}))$, ..., $F_{m,M} = \sigma(F_{m,(M-1)}\cup\sigma(j_{m,M}))$.

### E.1 PROOF OF LEMMA 1

We first prove Lemma 1, which is useful for step 4 in the main proof in Theorem 2 provided in Section E.2.

*Proof.* Recall the definition of the bias term: $\xi_n(\theta) = (\theta - \theta^*)^\top(g_n(\theta) - g(\theta))$. We have

$$\mathbb{E}\left[\xi_n(\theta)|\mathcal{F}_{n,0}\right]$$

$$= \mathbb{E}\left[(\theta - \theta^*)^\top(g_n(\theta) - g(\theta))|\mathcal{F}_{n,0}\right]$$

$$= \mathbb{E}\left[(\theta - \theta^*)^\top\left[\left(\frac{1}{M}\sum_{i=(n-1)M}^{nM-1}A_{x_i} - A\right)\theta + \left(\frac{1}{M}\sum_{i=(n-1)M}^{nM-1}b_{x_i} - b\right)\right]\bigg|\mathcal{F}_{n,0}\right]$$

$$\leq \frac{\lambda_A}{4}\mathbb{E}[\|\theta - \theta^*\|_2^2\,|\mathcal{F}_{n,0}] + \frac{1}{\lambda_A}\mathbb{E}\left[\left\|\left(\frac{1}{M}\sum_{i=(n-1)M}^{nM-1}A_{x_i} - A\right)\theta + \left(\frac{1}{M}\sum_{i=(n-1)M}^{nM-1}b_{x_i} - b\right)\right\|_2^2\bigg|\mathcal{F}_{n,0}\right]$$

$$\leq \frac{\lambda_A}{4}\mathbb{E}[\|\theta - \theta^*\|_2^2\,|\mathcal{F}_{n,0}] + \frac{2}{\lambda_A}\mathbb{E}\left[\left\|\left(\frac{1}{M}\sum_{i=(n-1)M}^{nM-1}A_{x_i} - A\right)\theta\right\|_2^2\bigg|\mathcal{F}_{n,0}\right]$$

$$+ \frac{2}{\lambda_A}\mathbb{E}\left[\left\|\frac{1}{M}\sum_{i=(n-1)M}^{nM-1}b_{x_i} - b\right\|_2^2\bigg|\mathcal{F}_{n,0}\right]$$

$$\leq \frac{\lambda_A}{4}\mathbb{E}[\|\theta - \theta^*\|_2^2\,|\mathcal{F}_{n,0}] + \frac{2R_\theta^2}{\lambda_A}\mathbb{E}\left[\left\|\frac{1}{M}\sum_{i=(n-1)M}^{nM-1}A_{x_i} - A\right\|_2^2\bigg|\mathcal{F}_{n,0}\right]$$

$$+ \frac{2}{\lambda_A}\mathbb{E}\left[\left\|\frac{1}{M}\sum_{i=(n-1)M}^{nM-1}b_{x_i} - b\right\|_2^2\bigg|\mathcal{F}_{n,0}\right]$$

$$\overset{(i)}{\leq} \frac{\lambda_A}{4}\mathbb{E}[\|\theta - \theta^*\|_2^2\,|\mathcal{F}_{n,0}] + \frac{2R_\theta^2}{\lambda_A}\mathbb{E}\left[\left\|\frac{1}{M}\sum_{i=(n-1)M}^{nM-1}A_{x_i} - A\right\|_F^2\bigg|\mathcal{F}_{n,0}\right]$$

$$+ \frac{2}{\lambda_A}\mathbb{E}\left[\left\|\frac{1}{M}\sum_{i=(n-1)M}^{nM-1}b_{x_i} - b\right\|_2^2\bigg|\mathcal{F}_{n,0}\right], \tag{20}$$

where $(i)$ follows from the fact that $\|W\|_2 \leq \|W\|_F$ for all matrices $W \in \mathbb{R}^{d\times d}$. We define the interproduct between two matices $W, V \in \mathbb{R}^{d\times d}$ as $\langle W, V\rangle = \sum_{ij}\sum_{ij}W_{ij}V_{ij}$. Consider the second term in eq. (20): $\mathbb{E}\left[\left\|\frac{1}{M}\sum_{i=(n-1)M}^{nM-1}A_{x_i} - A\right\|_F^2\big|\mathcal{F}_{n,0}\right]$, we have

$$\mathbb{E}\left[\left\|\frac{1}{M}\sum_{i=(n-1)M}^{nM-1}A_{x_i} - A\right\|_F^2\bigg|\mathcal{F}_{n,0}\right]$$

$$= \frac{1}{M^2}\sum_{i=(n-1)M}^{nM-1}\sum_{j=(n-1)M}^{nM-1}\mathbb{E}[\langle A_{x_i} - A, A_{x_j} - A\rangle|\mathcal{F}_{n,0}]$$

$$= \frac{1}{M^2}\left[\sum_{i=j}\mathbb{E}[\|A_{x_i} - A\|_F^2\,|\mathcal{F}_{n,0}] + \sum_{i\neq j}\mathbb{E}[\langle A_{x_i} - A, A_{x_j} - A\rangle|\mathcal{F}_{n,0}]\right]$$

$$\leq \frac{1}{M^2} \Big[ \sum_{i=j} \mathbb{E}[(\|A_{x_i}\|_F + \|A\|_F)^2 | \mathcal{F}_{n,0}] + \sum_{i \neq j} \mathbb{E}[\langle A_{x_i} - A, A_{x_j} - A \rangle | \mathcal{F}_{n,0}] \Big]$$

$$\leq \frac{1}{M^2} \Big[ 4(1+\gamma)^2 M + \sum_{i \neq j} \mathbb{E}[\langle A_{x_i} - A, A_{x_j} - A \rangle | \mathcal{F}_{n,0}]. \Big] \tag{21}$$

Now consider upper bound the term $\mathbb{E}[\langle A_{x_i} - A, A_{x_j} - A \rangle | \mathcal{F}_{n,0}]$ in eq. (21). Without loss of generality, we consider the case when $i > j$ as follows:

$$\mathbb{E}[\langle A_{x_i} - A, A_{x_j} - A \rangle | \mathcal{F}_{n,0}]$$
$$= \mathbb{E}\Big[ \mathbb{E}\big[ \langle A_{x_i} - A, A_{x_j} - A \rangle | x_j \big] \Big| \mathcal{F}_{n,0} \Big]$$
$$= \mathbb{E}\Big[ \langle \mathbb{E}\big[ A_{x_i} | x_j \big] - A, A_{x_j} - A \rangle \Big| \mathcal{F}_{n,0} \Big]$$
$$\leq \mathbb{E}\Big[ \big\| \mathbb{E}\big[ A_{x_i} | x_j \big] - A \big\|_F \big\| A_{x_j} - A \big\|_F \Big| \mathcal{F}_{n,0} \Big]$$
$$\leq \mathbb{E}\Big[ \big\| \mathbb{E}\big[ A_{x_i} | x_j \big] - A \big\|_F \big( \big\| A_{x_j} \big\|_F + \|A\|_F \big) \Big| \mathcal{F}_{n,0} \Big]$$
$$\leq 2(1+\gamma) \mathbb{E}\Big[ \big\| \mathbb{E}\big[ A_{x_i} | x_j \big] - A \big\|_F \Big| \mathcal{F}_{n,0} \Big]$$
$$\leq 2\kappa(1+\gamma)^2 \rho^{i-j}.$$

We can further obtain

$$\sum_{i \neq j} \mathbb{E}[\langle A_{x_i} - A, A_{x_j} - A \rangle | \mathcal{F}_{n,0}] \leq 2\kappa(1+\gamma)^2 \sum_{i \neq j} \rho^{|i-j|} \leq 2\kappa(1+\gamma)^2 \sum_{k=1}^{M-1} \sum_{l=1}^{k} \rho^l$$

$$\leq 2\kappa(1+\gamma)^2 \frac{2\rho}{1-\rho} \sum_{k=1}^{M-1} (1-\rho^k) \leq \frac{4(1+\gamma)^2 M \kappa \rho}{1-\rho}. \tag{22}$$

Then substituting eq. (22) into eq. (21) yields

$$\mathbb{E}\left[ \left\| \frac{1}{M} \sum_{i=(n-1)M}^{nM-1} A_{x_i} - A \right\|_F^2 \Bigg| \mathcal{F}_{n,0} \right] \leq \frac{1}{M} \Big[ 4(1+\gamma)^2 + \frac{4(1+\gamma)^2 \kappa \rho}{1-\rho} \Big].$$

Then consider the third term in eq. (20): $\mathbb{E}\left[ \left\| \frac{1}{M} \sum_{i=(n-1)M}^{nM-1} b_{x_i} - b \right\|_2^2 \Big| \mathcal{F}_{n,0} \right]$, similarly we have

$$\mathbb{E}\left[ \left\| \frac{1}{M} \sum_{i=(n-1)M}^{nM-1} b_{x_i} - b \right\|_2^2 \Bigg| \mathcal{F}_{n,0} \right]$$

$$= \frac{1}{M^2} \sum_{i=(n-1)M}^{nM-1} \sum_{j=(n-1)M}^{nM-1} \mathbb{E}[(b_{x_i} - b)^\top (b_{x_j} - b) | \mathcal{F}_{n,0}]$$

$$= \frac{1}{M^2} \Big[ \sum_{i=j} \mathbb{E}[\|b_{x_i} - b\|_2^2 | \mathcal{F}_{n,0}] + \sum_{i \neq j} \mathbb{E}[(b_{x_i} - b)^\top (b_{x_j} - b) | \mathcal{F}_{n,0}] \Big]$$

$$\leq \frac{1}{M^2} \Big[ \sum_{i=j} \mathbb{E}[(\|b_{x_i}\|_2 + \|b\|_2)^2 | \mathcal{F}_{n,0}] + \sum_{i \neq j} \mathbb{E}[(b_{x_i} - b)^\top (b_{x_j} - b) | \mathcal{F}_{n,0}] \Big]$$

$$\leq \frac{1}{M^2} \Big[ 4 r_{\max}^2 M + \sum_{i \neq j} \mathbb{E}[(b_{x_i} - b)^\top (b_{x_j} - b) | \mathcal{F}_{n,0}] \Big] \tag{23}$$

Now to upper-bound the term $\mathbb{E}[(b_{x_i} - b)^\top (b_{x_j} - b) | \mathcal{F}_{n,0}]$, without loss of generality, we assume $i > j$

$$\mathbb{E}[(b_{x_i} - b)^\top (b_{x_j} - b) | \mathcal{F}_{n,0}]$$

$$
\begin{aligned}
&= \mathbb{E}\Big[\mathbb{E}\big[(b_{x_i} - b)^\top (b_{x_j} - b)|x_j\big]\Big|\mathcal{F}_{n,0}\Big] \\
&= \mathbb{E}\Big[(\mathbb{E}\big[b_{x_i}|x_j\big] - b)^\top (b_{x_j} - b)\Big|\mathcal{F}_{n,0}\Big] \\
&\leq \mathbb{E}\Big[\big\|\mathbb{E}\big[b_{x_i}|x_j\big] - b\big\|_2 \big\|b_{x_j} - b\big\|_2 \Big|\mathcal{F}_{n,0}\Big] \\
&\leq \mathbb{E}\Big[\big\|\mathbb{E}\big[b_{x_i}|x_j\big] - b\big\|_2 \big(\big\|b_{x_j}\big\|_2 - \|b\|_2\big)\Big|\mathcal{F}_{n,0}\Big] \\
&\leq 2r_{\max}\mathbb{E}\Big[\big\|\mathbb{E}\big[b_{x_i}|x_j\big] - b\big\|_2 \Big|\mathcal{F}_{n,0}\Big] \\
&\leq 2\kappa r_{\max}^2 \rho^{i-j}.
\end{aligned}
$$

Thus, we have

$$
\sum_{i \neq j} \mathbb{E}[(b_{x_i} - b)^\top (b_{x_j} - b)|\mathcal{F}_{n,0}] \leq 2\kappa r_{\max}^2 \sum_{i \neq j} \rho^{|i-j|} \leq 2\kappa r_{\max}^2 \sum_{k=1}^{M-1}\sum_{l=1}^{k} \rho^l
$$

$$
\leq 2\kappa r_{\max}^2 \frac{2\rho}{1-\rho}\sum_{k=1}^{M-1}(1-\rho^k) \leq \frac{4r_{\max}^2 M\kappa\rho}{1-\rho}. \tag{24}
$$

Combing all of the above pieces together yields the following final bound

$$
\mathbb{E}\left[\xi_n(\theta)|\,\mathcal{F}_{n,0}\right] \leq \frac{\lambda_A}{4}\mathbb{E}[\|\theta - \theta^*\|_2^2 \,|\mathcal{F}_{n,0}] + \frac{8[1 + (\kappa - 1)\rho]}{\lambda_A(1-\rho)M}[R_\theta^2(1+\gamma)^2 + r_{\max}^2]. \tag{25}
$$

$\square$

## E.2  Proof of Theorem 2

**Step 1: Iteration within the $m$-th inner loop**

For the $m$-th inner loop, we consider the last update (i.e., the $M$-th iteration in the epoch), and decompose its error into the following form.

$$
\begin{aligned}
\|\theta_{m,M} - \theta^*\|_2^2 &= \left\|\Pi_{R_\theta}\Big(\theta_{m,M-1} + \alpha\big(g_{x_{j_{m,M}}}(\theta_{m,M-1}) - g_{x_{j_{m,M}}}(\tilde{\theta}_{m-1}) + g_m(\tilde{\theta}_{m-1})\big)\Big) - \theta^*\right\|_2^2 \\
&\leq \left\|\theta_{m,M-1} + \alpha\big(g_{x_{j_{m,M}}}(\theta_{m,M-1}) - g_{x_{j_{m,M}}}(\tilde{\theta}_{m-1}) + g_m(\tilde{\theta}_{m-1})\big) - \theta^*\right\|_2^2 \\
&= \|\theta_{m,M-1} - \theta^*\|_2^2 + 2\alpha(\theta_{m,M-1} - \theta^*)^\top\Big(g_{x_{j_{m,M}}}(\theta_{m,M-1}) - g_{x_{j_{m,M}}}(\tilde{\theta}_{m-1}) + g_m(\tilde{\theta}_{m-1})\Big) \\
&\quad + \alpha^2\left\|g_{x_{j_{m,M}}}(\theta_{m,M-1}) - g_{x_{j_{m,M}}}(\tilde{\theta}_{m-1}) + g_m(\tilde{\theta}_{m-1})\right\|_2^2. \tag{26}
\end{aligned}
$$

First, consider the third term in the right-hand side of eq. (26).

$$
\begin{aligned}
&\left\|g_{x_{j_{m,M}}}(\theta_{m,M-1}) - g_{x_{j_{m,M}}}(\tilde{\theta}_{m-1}) + g_m(\tilde{\theta}_{m-1})\right\|_2^2 \\
&= \left\|g_{x_{j_{m,M}}}(\theta_{m,M-1}) - g_{x_{j_{m,M}}}(\theta^*) - \Big[\big(g_{x_{j_{m,M}}}(\tilde{\theta}_{m-1}) - g_{x_{j_{m,M}}}(\theta^*)\big) - \big(g_m(\tilde{\theta}_{m-1}) - g_m(\theta^*)\big)\Big] + g_m(\theta^*)\right\|_2^2 \\
&\leq 3\left\|g_{x_{j_{m,M}}}(\theta_{m,M-1}) - g_{x_{j_{m,M}}}(\theta^*)\right\|_2^2 + 3\left\|\big(g_{x_{j_{m,M}}}(\tilde{\theta}_{m-1}) - g_{x_{j_{m,M}}}(\theta^*)\big) - \big(g_m(\tilde{\theta}_{m-1}) - g_m(\theta^*)\big)\right\|_2^2 \\
&\quad + 3\|g_m(\theta^*)\|_2^2. \tag{27}
\end{aligned}
$$

Then, by taking the expectation conditioned on $F_{m,(M-1)}$ on both sides of eq. (27), we have

$$
\begin{aligned}
&\mathbb{E}\left[\left\|g_{x_{j_{m,M}}}(\theta_{m,M-1}) - g_{x_{j_{m,M}}}(\tilde{\theta}_{m-1}) + g_m(\tilde{\theta}_{m-1})\right\|_2^2 \Big|F_{m,M-1}\right] \\
&\overset{(i)}{\leq} 3\mathbb{E}\left[\left\|g_{x_{j_{m,M}}}(\theta_{m,M-1}) - g_{x_{j_{m,M}}}(\theta^*)\right\|_2^2 \Big|F_{m,M-1}\right] \\
&\quad + 3\mathbb{E}\left[\left\|\big(g_{x_{j_{m,M}}}(\tilde{\theta}_{m-1}) - g_{x_{j_{m,M}}}(\theta^*)\big) - \mathbb{E}\big[g_{x_{j_{m,M}}}(\tilde{\theta}_{m-1}) - g_{x_{j_{m,M}}}(\theta^*)|F_{m,M-1}\big]\right\|_2^2 \Big|F_{m,M-1}\right]
\end{aligned}
$$

$$+ 3\mathbb{E}\left[\|g_m(\theta^*)\|_2^2 \,\Big|\, F_{m,M-1}\right]$$

$$\leq 3\mathbb{E}\left[\left\|g_{x_{j_{m,M}}}(\theta_{m,M-1}) - g_{x_{j_{m,M}}}(\theta^*)\right\|_2^2 \,\Big|\, F_{m,M-1}\right] + 3\mathbb{E}\left[\left\|g_{x_{j_{m,M}}}(\tilde{\theta}_{m-1}) - g_{x_{j_{m,M}}}(\theta^*)\right\|_2^2 \,\Big|\, F_{m,M-1}\right]$$

$$\overset{(ii)}{\leq} 3\mathbb{E}\left[\|A_{m,M}\|_2^2 \|\theta_{m,M-1} - \theta^*\|_2^2 \,\Big|\, F_{m,M-1}\right] + 3\mathbb{E}\left[\|A_{m,M}\|_2^2 \left\|\tilde{\theta}_{m-1} - \theta^*\right\|_2^2 \,\Big|\, F_{m,M-1}\right]$$

$$+ 3\mathbb{E}\left[\|g_m(\theta^*)\|_2^2 \,\Big|\, F_{1,0}\right]$$

$$\overset{(iii)}{\leq} 3(1+\gamma)^2 \|\theta_{m,M-1} - \theta^*\|_2^2 + 3(1+\gamma)^2 \left\|\tilde{\theta}_{m-1} - \theta^*\right\|_2^2 + 3\mathbb{E}\left[\|g_m(\theta^*)\|_2^2 \,\Big|\, F_{1,0}\right] \tag{28}$$

where $(i)$ follows from the fact that $\mathbb{E}[(g_{x_{j_{m,M}}}(\tilde{\theta}_{m-1}) - g_{x_{j_{m,M}}}(\theta^*))|F_{m,M-1}] = g_m(\tilde{\theta}_{m-1}) - g_m(\theta^*)$, $(ii)$ follows from the inequality $\mathbb{E}[(X - \mathbb{E}X)^2] \leq \mathbb{E}X^2$, and $(iii)$ follows from Lemma 2. We further consider the last term in eq. (28):

$$\mathbb{E}\left[\|g_m(\theta^*)\|_2^2 \,\Big|\, F_{1,0}\right] = \left\|\left(\frac{1}{M}\sum_{i=(m-1)M}^{mM-1} A_i\right)\theta^* + \left(\frac{1}{M}\sum_{i=(m-1)M}^{mM-1} b_i\right)\right\|_2^2.$$

Then, taking the expectation conditioned on $F_{m,M-1}$ on both sides of eq. (26) yields

$$\mathbb{E}\left[\|\theta_{m,M} - \theta^*\|_2^2 \,\Big|\, F_{m,M-1}\right]$$

$$\leq \|\theta_{m,M-1} - \theta^*\|_2^2 + 2\alpha(\theta_{m,M-1} - \theta^*)^\top \mathbb{E}\left[g_{x_{j_{m,M}}}(\theta_{m,M-1}) - g_{x_{j_{m,M}}}(\tilde{\theta}_{m-1}) + g_m(\tilde{\theta}_{m-1}) \,\Big|\, F_{m,M-1}\right]$$

$$+ \alpha^2 \mathbb{E}\left[\left\|g_{x_{j_{m,M}}}(\theta_{m,M-1}) - g_{x_{j_{m,M}}}(\tilde{\theta}_{m-1}) + \tilde{g}_m\right\|_2^2 \,\Big|\, F_{m,M-1}\right]$$

$$\overset{(i)}{\leq} \|\theta_{m,M-1} - \theta^*\|_2^2 + 2\alpha(\theta_{m,M-1} - \theta^*)^\top \mathbb{E}\left[g_{x_{j_{m,M}}}(\theta_{m,M-1}) \,\Big|\, F_{m,M-1}\right] + 3\alpha^2(1+\gamma)^2 \|\theta_{m,M-1} - \theta^*\|_2^2$$

$$+ 3\alpha^2(1+\gamma)^2 \left\|\tilde{\theta}_{m-1} - \theta^*\right\|_2^2 + 3\alpha^2 \mathbb{E}\left[\|g_m(\theta^*)\|_2^2 \,\Big|\, F_{1,0}\right]$$

$$\overset{(ii)}{=} \|\theta_{m,M-1} - \theta^*\|_2^2 + 2\alpha(\theta_{m,M-1} - \theta^*)^\top g(\theta_{m,M-1}) + 2\alpha\mathbb{E}\left[\xi_m(\theta_{m,M-1}) \,\Big|\, F_{m,M-1}\right]$$

$$+ 3\alpha^2(1+\gamma)^2 \|\theta_{m,M-1} - \theta^*\|_2^2 + 3\alpha^2(1+\gamma)^2 \left\|\tilde{\theta}_{m-1} - \theta^*\right\|_2^2 + 3\alpha^2 \mathbb{E}\left[\|g_m(\theta^*)\|_2^2 \,\Big|\, F_{1,0}\right]$$

$$\leq \|\theta_{m,M-1} - \theta^*\|_2^2 - [\alpha\lambda_A - 3\alpha^2(1+\gamma)^2] \|\theta_{m,M-1} - \theta^*\|_2^2 + 3\alpha^2(1+\gamma)^2 \left\|\tilde{\theta}_{m-1} - \theta^*\right\|_2^2$$

$$+ 2\alpha\mathbb{E}\left[\xi_m(\theta_{m,M-1}) \,\Big|\, F_{m,M-1}\right] + 3\alpha^2 \mathbb{E}\left[\|g_m(\theta^*)\|_2^2 \,\Big|\, F_{1,0}\right], \tag{29}$$

where $(i)$ follows by plugging eq. (28) into its preceding step and from the fact that $\mathbb{E}\left[g_{x_{j_{m,M}}}(\tilde{\theta}_{m-1}) - g_m(\tilde{\theta}_{m-1}) \,\Big|\, F_{m,M-1}\right] = 0$. In $(ii)$ we define $\xi_m(\theta) = (\theta - \theta^*)^\top (g_m(\theta) - g(\theta))$ for $\theta \in \mathbb{R}^d$. Then, by applying eq. (29) iteratively, we have

$$\mathbb{E}\left[\|\theta_{m,1} - \theta^*\|_2^2 \,\Big|\, F_{m,0}\right]$$

$$\leq \|\theta_{m,0} - \theta^*\|_2^2 - [\alpha\lambda_A - 3\alpha^2(1+\gamma)^2] \sum_{i=0}^{M-1} \mathbb{E}\left[\|\theta_{m,i} - \theta^*\|_2^2 \,\Big|\, F_{m,0}\right] + 3M\alpha^2(1+\gamma)^2 \left\|\tilde{\theta}_{m-1} - \theta^*\right\|_2^2$$

$$+ 2\alpha \sum_{i=0}^{M-1} \mathbb{E}\left[\xi_m(\theta_{m,i}) \,\Big|\, F_{m,0}\right] + 3M\alpha^2 \mathbb{E}\left[\|g_m(\theta^*)\|_2^2 \,\Big|\, F_{1,0}\right]. \tag{30}$$

Arranging the terms in eq. (30) yields

$$[\alpha\lambda_A - 3\alpha^2(1+\gamma)^2] \sum_{i=0}^{M-1} \mathbb{E}\left[\|\theta_{m,i} - \theta^*\|_2^2 \,\Big|\, F_{m,0}\right]$$

$$\leq [1 + 3M\alpha^2(1+\gamma)^2] \left\|\tilde\theta_{m-1} - \theta^*\right\|_2^2 + 2\alpha \sum_{i=0}^{M-1} \mathbb{E}\left[\xi_m(\theta_{m,i}) \Big| F_{m,0}\right] + 3M\alpha^2 \mathbb{E}\left[\|g_m(\theta^*)\|_2^2 \Big| F_{1,0}\right].$$
(31)

Then, substituting eq. (25) into eq. (31), we obtain

$$[\alpha\lambda_A - 3\alpha^2(1+\gamma)^2] \sum_{i=0}^{M-1} \mathbb{E}\left[\|\theta_{m,i} - \theta^*\|_2^2 \Big| F_{m,0}\right]$$

$$\leq [1 + 3M\alpha^2(1+\gamma)^2] \left\|\tilde\theta_{m-1} - \theta^*\right\|_2^2 + \frac{\lambda_A\alpha}{2} \sum_{i=0}^{M-1} \mathbb{E}\left[\|\theta_{m,i} - \theta^*\|_2^2 | F_{m,0}\right]$$

$$+ \frac{16[1 + (\kappa-1)\rho]\alpha}{\lambda_A(1-\rho)}[R_\theta^2(1+\gamma)^2 + r_{\max}^2] + 3M\alpha^2 \mathbb{E}\left[\|g_m(\theta^*)\|_2^2 \Big| F_{1,0}\right].$$
(32)

Subtracting $0.5\lambda_A\alpha \sum_{i=0}^{M-1} \mathbb{E}[\|\theta_{m,i} - \theta^*\|_2^2 | F_{m,0}]$ on both sides of eq. (32) yields

$$[0.5\alpha\lambda_A - 3\alpha^2(1+\gamma)^2] \sum_{i=0}^{M-1} \mathbb{E}\left[\|\theta_{m,i} - \theta^*\|_2^2 \Big| F_{m,0}\right]$$

$$\leq [1 + 3M\alpha^2(1+\gamma)^2] \left\|\tilde\theta_{m-1} - \theta^*\right\|_2^2 + \frac{16[1 + (\kappa-1)\rho]\alpha}{\lambda_A(1-\rho)}[R_\theta^2(1+\gamma)^2 + r_{\max}^2]$$

$$+ 3M\alpha^2 \mathbb{E}\left[\|g_m(\theta^*)\|_2^2 \Big| F_{1,0}\right].$$
(33)

Then, dividing eq. (31) by $[0.5\alpha\lambda_A - 3\alpha^2(1+\gamma)^2]M$ on both sides, we obtain

$$\mathbb{E}\left[\left\|\tilde\theta_m - \theta^*\right\|_2^2 \Big| F_{m,0}\right]$$

$$\leq \frac{1/M + 3\alpha^2(1+\gamma)^2}{0.5\alpha\lambda_A - 3\alpha^2(1+\gamma)^2} \left\|\tilde\theta_{m-1} - \theta^*\right\|_2^2 + \frac{16[1 + (\kappa-1)\rho][R_\theta^2(1+\gamma)^2 + r_{\max}^2]\alpha}{\lambda_A(1-\rho)[0.5\alpha\lambda_A - 3\alpha^2(1+\gamma)^2]M}$$

$$+ \frac{3\alpha}{0.5\lambda_A - 3\alpha(1+\gamma)^2} \mathbb{E}\left[\|g_m(\theta^*)\|_2^2 \Big| F_{1,0}\right].$$
(34)

For simplicity, let $C_1 = \frac{1/M + 3\alpha^2(1+\gamma)^2}{0.5\alpha\lambda_A - 3\alpha^2(1+\gamma)^2}$, $C_2 = \frac{16[1 + (\kappa-1)\rho][R_\theta^2(1+\gamma)^2 + r_{\max}^2]}{1-\rho}$ and $C_3 = \frac{3\alpha}{0.5\lambda_A - 3\alpha(1+\gamma)^2}$. Then we rewrite eq. (34):

$$\mathbb{E}\left[\left\|\tilde\theta_m - \theta^*\right\|_2^2 \Big| F_{m,0}\right] \leq C_1 \left\|\tilde\theta_{m-1} - \theta^*\right\|_2^2 + \frac{C_2}{[0.5\lambda_A - 3\alpha(1+\gamma)^2]\lambda_A M} + C_3 \mathbb{E}\left[\|g_m(\theta^*)\|_2^2 \Big| F_{1,0}\right].$$
(35)

**Step 2: Iteration over $m$ epochs**

Taking the expectation of eq. (35) conditioned on $F_{m-1,0}$ and upper-bounding $\mathbb{E}\left[\left\|\tilde\theta_{m-1} - \theta^*\right\|_2^2\right]$ by following similar steps in the previous steps, we obtained

$$\mathbb{E}\left[\left\|\tilde\theta_m - \theta^*\right\|_2^2 \Big| F_{m-1,0}\right]$$

$$\leq C_1 \left\|\tilde\theta_{m-1} - \theta^*\right\|_2^2 + \frac{C_2}{[0.5\lambda_A - 3\alpha(1+\gamma)^2]\lambda_A M} + C_3 \mathbb{E}\left[\|g_m(\theta^*)\|_2^2 \Big| F_{1,0}\right]$$

$$\leq C_1^2 \left\|\tilde\theta_{m-2} - \theta^*\right\|_2^2 + \frac{C_2}{[0.5\lambda_A - 3\alpha(1+\gamma)^2]\lambda_A M} \sum_{k=0}^{1} C_1^k + C_3 \sum_{k=0}^{1} C_1^k \mathbb{E}\left[\|g_{m-k}(\theta^*)\|_2^2 \Big| F_{1,0}\right].$$

By following the above steps for $(m-1)$ times, we have

$$\mathbb{E}\left[\left\|\tilde\theta_m - \theta^*\right\|_2^2 \Big| F_{1,0}\right]$$

$$\leq C_1^m \left\|\tilde{\theta}_0 - \theta^*\right\|_2^2 + \frac{C_2}{[0.5\lambda_A - 3\alpha(1+\gamma)^2]\lambda_A M} \sum_{k=0}^{m-1} C_1^k + C_3 \sum_{k=0}^{m-1} C_1^k \mathbb{E}\left[\|g_{m-k}(\theta^*)\|_2^2 \Big| F_{1,0}\right].$$

(36)

Then taking the expectation of $\sigma(S)$ (which contains the randomness of the entire sample trajectory) on both sides of eq. (36) yields

$$\mathbb{E}\left[\left\|\tilde{\theta}_m - \theta^*\right\|_2^2\right]$$

$$\leq C_1^m \left\|\tilde{\theta}_0 - \theta^*\right\|_2^2 + \frac{C_2}{[0.5\lambda_A - 3\alpha(1+\gamma)^2]\lambda_A M} \sum_{k=0}^{m-1} C_1^k + C_3 \sum_{k=0}^{m-1} C_1^k \mathbb{E}\left[\|g_{m-k}(\theta^*)\|_2^2\right],$$

(37)

where the second term in the right hand side of eq. (37) corresponds to the bias error and the third term corresponds to the variance error.

**Step 3: Bounding the variance error**

For any $0 \leq k \leq m-1$, we have

$$\|g_{m-k}(\theta^*)\|_2^2 = \left\|\frac{1}{M} \sum_{i=(m-k-1)M}^{(m-k)M-1} g_{x_i}(\theta^*)\right\|_2^2$$

$$= \frac{1}{M^2} \left(\sum_{i=(m-k-1)M}^{(m-k)M-1} g_{x_i}^\top(\theta^*)\right)\left(\sum_{j=(m-k-1)M}^{(m-k)M-1} g_{x_j}(\theta^*)\right)$$

$$= \frac{1}{M^2} \sum_{i=(m-k-1)M}^{(m-k)M-1} \sum_{j=(m-k-1)M}^{(m-k)M-1} g_{x_i}^\top(\theta^*)g_{x_j}(\theta^*)$$

$$= \frac{1}{M^2} \sum_{i=j} \|g_{x_i}(\theta^*)\|_2^2 + \frac{1}{M^2} \sum_{i\neq j} g_{x_i}^\top(\theta^*)g_{x_j}(\theta^*)$$

$$\overset{(i)}{\leq} \frac{G^2}{M} + \frac{1}{M^2} \sum_{i\neq j} g_{x_i}^\top(\theta^*)g_{x_j}(\theta^*),$$

(38)

where $(i)$ follows from Lemma 3. Consider the expectation of the second term in eq. (38), which is given by

$$\frac{1}{M^2} \sum_{i\neq j} \mathbb{E}[g_{x_i}^\top(\theta^*)g_{x_j}(\theta^*)].$$

(39)

Without loss of generality, we consider the case when $j > i$ as follows:

$$\mathbb{E}[g_{x_i}^\top(\theta^*)g_{x_j}(\theta^*)] = \mathbb{E}[\mathbb{E}[g_{x_j}(\theta^*)|P_i]^\top g_{x_i}(\theta^*)]$$

$$\leq \mathbb{E}[\left\|\mathbb{E}[g_{x_j}(\theta^*)|P_i]\right\|_2 \|g_{x_i}(\theta^*)\|_2]$$

$$\leq G\mathbb{E}[\left\|\mathbb{E}[g_{x_j}(\theta^*)|P_i]\right\|_2]$$

$$= G\mathbb{E}[\left\|\mathbb{E}[(A_j\theta^* + b_j)|P_i]\right\|_2]$$

$$\leq G\mathbb{E}[\left\|\mathbb{E}[A_j|P_i]\theta^* + \mathbb{E}[b_j|P_i]\right\|_2]$$

$$= G\mathbb{E}[\left\|(\mathbb{E}[A_j|P_i] - A)\theta^* + (\mathbb{E}[b_j|P_i] - b)\right\|_2]$$

$$\leq G\mathbb{E}[\left\|(\mathbb{E}[A_j|P_i] - A)\theta^*\right\|_2 + \left\|\mathbb{E}[b_j|P_i] - b\right\|_2]$$

$$\leq G\mathbb{E}[\left\|\mathbb{E}[A_j|P_i] - A\right\|_2 \|\theta^*\|_2 + \left\|\mathbb{E}[b_j|P_i] - b\right\|_2]$$

$$\leq \kappa G[(1+\gamma)R_\theta + r_{\max}]\rho^{j-i}.$$

(40)

Substituting eq. (40) into eq. (39), we obtain

$$\frac{1}{M^2} \sum_{i\neq j} \mathbb{E}[g_{x_i}^\top(\theta^*)g_{x_j}(\theta^*)] \leq \frac{\kappa G[(1+\gamma)R_\theta + r_{\max}]}{M^2} \sum_{i\neq j} \rho^{|i-j|}$$

$$\leq \frac{\kappa G[(1+\gamma)R_\theta + r_{\max}]}{M^2} \left(2M \sum_{k=1}^{\lceil \frac{M}{2} \rceil} \rho^k \right)$$

$$\leq \frac{2\rho\kappa G[(1+\gamma)R_\theta + r_{\max}]}{(1-\rho)M}. \tag{41}$$

Then substituting eq. (41) into eq. (38) yields

$$\mathbb{E}[\|g_{m-k}(\theta^*)\|_2^2] \leq \frac{1}{M}\left(G^2 + \frac{2\rho\kappa G[(1+\gamma)R_\theta + r_{\max}]}{(1-\rho)}\right) \leq \frac{C_4}{M}, \tag{42}$$

where $C_4 = G^2 + \frac{2\rho\kappa G[(1+\gamma)R_\theta + r_{\max}]}{(1-\rho)}$. Finally, substituting eq. (42) into the accumulated residual variance term in eq. (37), we have

$$C_3 \sum_{k=0}^{m-1} C_1^k \mathbb{E}\left[\|g_{m-k}(\theta^*)\|_2^2\right] \leq \frac{C_3 C_4}{M} \sum_{k=0}^{m-1} C_1^k \leq \frac{C_3 C_4}{(1-C_1)M}. \tag{43}$$

**Step 4: Combining all error terms**

Finally, substituting eq. (43) and substituting the values of $C_2$ and $C_3$ into eq. (37), we have

$$\mathbb{E}\left[\left\|\tilde{\theta}_m - \theta^*\right\|_2^2\right] \leq C_1^m \left\|\tilde{\theta}_0 - \theta^*\right\|_2^2 + \frac{3C_4\alpha + C_2/\lambda_A}{(1-C_1)[0.5\lambda_A - 3\alpha(1+\gamma)^2]M},$$

which yields the desired result.

## F   SAMPLE COMPLEXITY OF TD

The finite-time convergence rate of vanilla TD under i.i.d. and Markovian sampling has been characterized in Bhandari et al. (2018); Srikant and Ying (2019). However, these studies did not provide the overall computational complexity, i.e., the total number of pseudo-gradient computations to achieve an $\epsilon$-accuracy solution. This section provides such an analysis based on their convergence results for completeness.

### F.1   TD WITH I.I.D. SAMPLES

Consider the vanilla TD update in Bhandari et al. (2018). Following the steps similar to those in Bhandari et al. (2018) for proving Theorem 2, and letting the constant stepsize $\alpha \leq \min\{\frac{\lambda_A}{4(1+\gamma)^2}, \frac{2}{\lambda_A}\}$, we have

$$\mathbb{E}\|\theta_t - \theta^*\|_2^2 \leq (1 - \frac{1}{2}\lambda_A\alpha)^t \|\theta_0 - \theta^*\|_2^2 + \frac{2C_5\alpha}{\lambda_A}$$

$$\leq e^{-\frac{1}{2}\lambda_A\alpha t} \|\theta_0 - \theta^*\|_2^2 + \frac{2C_5\alpha}{\lambda_A},$$

where $0 < C_5 < \infty$ is a constant. Let $\alpha = \min\{\frac{\lambda_A}{4(1+\gamma)^2}, \frac{2}{\lambda_A}, \frac{\epsilon\lambda_A}{4C_5}\}$. It can be checked easily that with the total number of iterations at most

$$t = \lceil \frac{2}{\lambda_A\alpha} \log(\frac{2\|\theta_0 - \theta^*\|_2^2}{\epsilon}) \rceil = \lceil 2\max\{\frac{4(1+\gamma)^2}{\lambda_A^2}, 1, \frac{4C_5}{\epsilon\lambda_A^2}\} \log(\frac{2\|\theta_0 - \theta^*\|_2^2}{\epsilon}) \rceil$$

$$= \mathcal{O}\left(\left(\frac{1}{\epsilon\lambda_A^2}\right) \log\left(\frac{1}{\epsilon}\right)\right), \tag{44}$$

an $\epsilon$-accurate solution can be attained, i.e., $\mathbb{E}\|\theta_t - \theta^*\|_2^2 \leq \epsilon$. Since each iteration requires one pseudo-gradient computation, the total number of pseudo-gradient computations is also given by eq. (44).

## F.2 TD with Markovian samples

Consider the vanilla TD update in Bhandari et al. (2018). Following the steps similar to those in Bhandari et al. (2018) for proving Theorem 3, and letting the constant stepsize $\alpha \leq \frac{1}{\lambda_A}$, we have

$$\mathbb{E} \|\theta_t - \theta^*\|_2^2 \leq (1 - \lambda_A \alpha)^t \|\theta_0 - \theta^*\|_2^2 + \frac{C_6 \alpha}{\lambda_A} + \frac{C_7 \alpha \log(\frac{1}{\alpha})}{\lambda_A}$$

$$\leq e^{-\lambda_A \alpha t} \|\theta_0 - \theta^*\|_2^2 + \frac{C_6 \alpha}{\lambda_A} + \frac{C_7 \alpha \log(\frac{1}{\alpha})}{\lambda_A}$$

where $0 < C_6 < \infty$ and $0 < C_7 < \infty$ are constants. Now let $\alpha = \min\{\frac{C_8 \epsilon}{\log(1/C_8 \epsilon)}, \frac{1}{\lambda_A}\}$ where $C_8 = \lambda_A \min\{\frac{1}{C_6}, \frac{1}{6C_7}\}$, it can be checked easily that with the total number of iterations at most

$$t = \lceil \frac{2}{\lambda_A \alpha} \log(\frac{3 \|\theta_0 - \theta^*\|_2^2}{\epsilon}) \rceil$$

$$= \lceil \max\{\frac{2}{\min\{\frac{1}{C_6}, \frac{1}{6C_7}\}\lambda_A^2 \epsilon}, 2\} \log\left(\frac{1}{C_8 \epsilon}\right) \log(\frac{3 \|\theta_0 - \theta^*\|_2^2}{\epsilon}) \rceil$$

$$= \mathcal{O}\left(\left(\frac{1}{\epsilon \lambda_A^2}\right) \log^2\left(\frac{1}{\epsilon}\right)\right), \tag{45}$$

an $\epsilon$-accurate solution can be attained, i.e., $\mathbb{E} \|\theta_t - \theta^*\|_2^2 \leq \epsilon$. Since each iteration requires one pseudo-gradient computation, the total number of pseudo-gradient computations is also given by eq. (45).

