# OpenReview forum: "Reanalysis of Variance Reduced Temporal Difference Learning"
_ICLR.cc/2020/Conference — Accept (Poster)_

### Official Review · AnonReviewer1 · 2019-10-21
**Official Blind Review #1**

**Rating:** 6

**Review:**

Summary:
In this paper, the authors study the variance reduced TD (VRTD) algorithm, by Korda and Prashanth (2015)  (KP15), for policy evaluation in RL. They first highlight technical errors in the analysis of KP15, and then provide new convergence analysis for this algorithm. The new analysis is based on a new technique to bound the bias of the VRTD gradient estimator, and shows the advantage of VRTD over vanilla TD (the analyses by Bhandari et al. 2018 and Srikant and Ying 2019), both in terms of variance and bias, that are reduced by increasing the batch size. The authors show that while the variance and bias of vanilla TD are both of O(\alpha) (where \alpha is the step-size), they are of O(\alpha/M) and O(1/\sqrt{M}) (where M is the batch size) in VRTD. This shows that a good convergence is possible for VRTD without reducing the step-size \alpha, that causes slow convergence, by increasing the batch size M. In the middle of their analysis, the authors propose a slight modification of VRTD for the case that the samples are obtained iid from the stationary distribution of the evaluating policy, and provide its analysis. Finally, the authors provide simple experiments to support their theoretical findings.

Comments:
- More discussion on the parameter C_0, a constant between 0 and \infty that depends on the MDP, is necessary in the paper. The authors provide no discussion about this constant and only in the appendix, refer the readers to Dedecker and Gouezel (2015). This constant is important because if it is large, then the batch size M should be very big in order for the bias error to go to zero. It would be good to see that which MDP properties affect the value of C_0.
- As we increase M, the final performance gets better, but the (sample) cost of each gradient update increases. It would be good to have a comparison between different M values, in terms of performance vs. number of samples. What I mean is to have figures similar to 1(a) and 1(b) in which the performance is on the y-axis and the number of samples on the x-axis.
- I was wondering if we can get the same improvement in terms of performance by having a decreasing schedule for \alpha. What I mean is instead of increasing M, we keep M constant and then define a decreasing schedule for \alpha and prove similar bounds that go to zero as \alpha goes to zero.

Minor Comments:
- The projection \Pi_{R_\theta} in Algorithm 2 has not been explained. This is important as R_\theta appears in some of the theoretical results later in the paper.
- \lambda_A  has been used on Page 5 without definition. The authors define this quantity later on Page 6.
- What is \Psi in the discussions at the bottom of Page 5 (Eqs. 2, 3, and 4). Is it \Phi? Overall, I think this discussion would be more meaningful if the authors first introduce the terms used in Eqs. 2 and 3.


**Experience Assessment:**

I have published one or two papers in this area.

**Review Assessment: Checking Correctness Of Derivations And Theory:**

I assessed the sensibility of the derivations and theory.

**Review Assessment: Checking Correctness Of Experiments:**

I assessed the sensibility of the experiments.

**Review Assessment: Thoroughness In Paper Reading:**

I read the paper at least twice and used my best judgement in assessing the paper.

---

> ### Author Response · Authors · 2019-11-14
> **We thank the reviewer very much for the very helpful comments! Below is a point-to-point response.**
>
> Q1: More discussion on the parameter C_0, a constant between 0 and \infty that depends on the MDP, is necessary in the paper. The authors provide no discussion about this constant and only in the appendix, refer the readers to Dedecker and Gouezel (2015). This constant is important because if it is large, then the batch size M should be very big in order for the bias error to go to zero. It would be good to see that which MDP properties affect the value of C_0.
>
> A: Thank you for the suggestion! C_0 depends proportionally on the coupling time and the returning time of the underlying Markov chain. Specifically, the coupling time indicates how fast the Markov chain converges to the stationary distribution, and the returning time, which depends on the nature of the stationary distribution, captures the expected time that the Markov chain returns to the same state. The value of C_0 is typically small if the Markov chain has a small mixing time and the stationary distribution is less degenerate.
>
>
> Q2: As we increase M, the final performance gets better, but the (sample) cost of each gradient update increases. It would be good to have a comparison between different M values, in terms of performance vs. number of samples. What I mean is to have figures similar to 1(a) and 1(b) in which the performance is on the y-axis and the number of samples on the x-axis.
>
> A: We want to clarify that in both figures 1 (a, Left) and (b, Left), the x_axis corresponds to the number of gradient computations, which equivalently corresponds to the number of samples used. Our experiments show that the advantage of VRTD (compared to vanilla TD) mainly lies in the high accuracy regime, in which the sample complexity (i.e., gradient computational cost) is much lower for VRTD with larger batch size M.
>
>
> Q3: Whether we can get the same improvement in terms of performance by decreasing \alpha instead of increasing M, we keep M constant and then define a decreasing schedule for \alpha and prove similar bounds that go to zero as \alpha goes to zero.
>
> A: One can obtain a sample complexity result of VRTD under a diminishing stepsize \alpha and a fixed batch size M. In this case, we expect that the convergence rate will be sublinear (instead of the linear convergence under constant stepsize) due to the use of diminishing stepsize. Moreover, the bound can converge to zero if the scheduling of \alpha satisfies certain conditions such as \sum_t \alpha_t^2 = \infty. Such a result will be similar to that of standard SGD, because the large batch gradient sampled by VRTD is still an inexact estimation of the population gradient with reduced variance.
>
> Minor comments:
>
> Q4: The projection \Pi_{R_\theta} in Algorithm 2 has not been explained.
>
> A: \Pi_{R_\theta} denotes the projection operator onto a norm ball of radius R_\theta. We have clarified this in Section 3.1 in our revision.
>
>
> Q5: \lambda_A has been used on Page 5 without definition. The authors define this quantity later on Page 6.
>
> A: We have defined \lambda_A in page 5 when it is first mentioned in the revision.
>
>
> Q6: What is \Psi in the discussions at the bottom of Page 5 (Eqs. 2, 3, and 4). Is it \Phi? Overall, I think this discussion would be more meaningful if the authors first introduce the terms used in Eqs. 2 and 3.
>
> A: Thanks for pointing this out. \Psi denotes the stationary distribution of the corresponding Markov chain, and \Psi is not \Phi. We have added the definition for \Psi below eq (2) in the revision.

---

### Official Review · AnonReviewer3 · 2019-10-29
**Official Blind Review #3**

**Rating:** 3

**Review:**

The paper is on temporal difference learning, specifically variance reduction of it. As per the claims of the paper, a previous method from (Korda and La, 2015) had technical errors, which the paper corrects and provides a better analysis of variance reduction. In the end, the paper focuses on the variance of the gradient estimator in temporal difference learning, and on analysis of the bias error. The final method is validated on two experiments with iid and Markovian sampling.

Strengths:

+ The paper seems to conduct some serious analysis of the variance in temporal difference learning. The discussed analysis is substantial and the derivations seem to check out.

Weaknesses:

+ I believe the paper could improve its clarity for someone who is not an expert on reinforcement learning. In the current version I find it a bit hard to understand the direction the paper takes.

+ The experiments are quite thin and also vague. It is unclear what exactly is the experimental setup, how were the data generated and what are fair baselines. In fact, not even competitor methods or the method of (Korda and La, 2015) are included in the comparisons. As such, it is not possible to understand whether the proposed method is really contributing something or not. What would be the results in standard RL environments, like from the OpenAI gym or the MuJoCo environments?

I apologize for my rather short review. Unfortunately, I find it hard to write something more without going very deep on the derivations, which I could not give shortage of time. I believe the paper is of value and I am willing to upgrade my score upon a more persuasive experimentation.

**Experience Assessment:**

I have read many papers in this area.

**Review Assessment: Checking Correctness Of Derivations And Theory:**

I assessed the sensibility of the derivations and theory.

**Review Assessment: Checking Correctness Of Experiments:**

I carefully checked the experiments.

**Review Assessment: Thoroughness In Paper Reading:**

I read the paper at least twice and used my best judgement in assessing the paper.

---

> ### Author Response · Authors · 2019-11-14
> **We thank the reviewer very much for the very helpful comments! Below is a point-to-point response.**
>
> Q1: I believe the paper could improve its clarity for someone who is not an expert on reinforcement learning. In the current version I find it a bit hard to understand the direction the paper takes.
>
> A:  We will make our best efforts to improve the clarity of the paper in the revision.
>
>
> Q2: The experiments are quite thin and also vague. It is unclear what exactly is the experimental setup, how were the data generated and what are fair baselines. In fact, not even competitor methods or the method of (Korda and La, 2015) are included in the comparisons. As such, it is not possible to understand whether the proposed method is really contributing something or not. What would be the results in standard RL environments, like from the OpenAI gym or the MuJoCo environments?
>
> A: Thank you very much for the suggestions. We have added the details suggested by the reviewer about the experimental setup at the beginning of Section 5 in the revision. The baseline for comparison is the vanilla TD algorithm, which corresponds to the case with M=1 in our figure.
>
> Following the suggestion by the reviewer, in the updated paper, we have provided additional experiments in supplementary materials Appendix G, in which we studied two problems in OpenAI Gym: Frozen Lake and Mountain Car. Both experiments indicate that VRTD achieves a much smaller error than vanilla TD (i.e., M=1), and increasing the batch size for VRTD substantially reduces the error without much slowing down the convergence.

---

### Official Review · AnonReviewer2 · 2019-11-01
**Official Blind Review #2**

**Rating:** 6

**Review:**

This paper provides theoretical guarantees for variance reduced temporal difference algorithms. It points out an analysis error in a previous paper and gives the correct proof. The analyses are based on two sampling schemes, i.i.d. and Markovian.  Convergence guarantees and convergence rates are provided for both sampling schemes.

The paper is well written. The main theoretical results are presented clearly. The convergence rates are reasonable to me. Considering the importance of TD, I think this analysis will provide some insights for future directions of TD.

I have several questions and concerns.
1. The data is simulated. I am wondering how VRTD works in practice comparing to naive TD.
2. The number of gradient evaluations is used in x-axis in the experimental results. What if it's replaced by wall-clock time?

**Experience Assessment:**

I do not know much about this area.

**Review Assessment: Checking Correctness Of Derivations And Theory:**

I assessed the sensibility of the derivations and theory.

**Review Assessment: Checking Correctness Of Experiments:**

I assessed the sensibility of the experiments.

**Review Assessment: Thoroughness In Paper Reading:**

I made a quick assessment of this paper.

---

> ### Author Response · Authors · 2019-11-14
> **We thank the reviewer very much for the very helpful comments! Below is a point-to-point response.**
>
> Q1: The data is simulated. I am wondering how VRTD works in practice compared to naive TD.
>
> A: In the updated paper, we have provided additional experiments in supplementary materials Appendix G, in which we studied two problems in OpenAI Gym: Frozen Lake and Mountain Car. Both experiments indicate that VRTD achieves a much smaller error than vanilla TD (i.e., M=1), and increasing the batch size for VRTD substantially reduces the error without much slowing down the convergence.
>
>
> Q2: The number of gradient evaluations is used in the x-axis in the experimental results. What if it's replaced by wall-clock time?
>
> A: As the wall-clock time for computing the gradient of a single sample is almost a universal constant (independent of the batch size, the progress of iteration, etc), the plot based on wall-clock time is the same as that based on the number of gradient evaluations.

---

### Official Review · AnonReviewer5 · 2019-11-03
**Official Blind Review #5**

**Rating:** 8

**Review:**

This paper analyzes a TD algorithm with batch estimation of the TD step with the purpose of variance reduction in both the iid and Markov noise setups. This work is brought as reanalysis of the centered TD algorithm from Korda and La (2015), which is known to contain several errors in its analysis and statements.

The contributions are in the form of two respective finite-time bounds for the iid and Markov noise, that exhibit improvement over vanilla TD by a factor of O(1/M) for the variance and O(1/\sqrt{M}) for the bias, at the expense of M inner iterations instead of 1. Even though it is usually challenging to convey this type of analysis in a short manuscript, this work makes it accessible and is well written. The counterexample to existing errors in previous proofs is compelling, and the preceding discussion regarding the proofs prior to each theorem make it easy to immediately catch the gist of the proofs without having to go through the appendix.

Despite the advantages above, I am disturbed by the lack of comparison of the computational burden introduced by the M inner loops to vanilla TD. It is not clear from the results whether a practitioner would prefer paying those extra computations to reduce the bias and variance by the M-dependent factors in the convergence rate. Second, the novelty of this work is not highly significant. It indeed corrects a bound for an existing algorithm, but not more than that. Namely, it only analyzes the so-called VRTD with a specific stepsize O(1/n), which also contains a system-dependent parameter that is not known to the user -- the smallest eigenvalue of the driving matrix.

While I don't believe the authors can do much to mitigate the second of the two qualms above, I am open to upgrade my score if they can provide a compelling answer to the first, and answer the questions bellow.

The following comments/questions are in order:
1. In several locations the TD update is referred to as a gradient. Despite the explanation as of why in Footnote 1, this is misleading. Since it is known that the TD update is *not* a true gradient step, it would be wiser not to refer to it as one.
2. Can't Algorithm 1 be easily improved to be more efficient? In step 7 you perform M updates but end up throwing away M-t of them, where t is drawn randomly (step 10). Instead, you can simply draw t at the beginning and compute step 7 only up to that t. Is that correct?
3. Theorem 1: mention what exactly is the M you refer to -- that might expose more interesting relations.
4. Paragraph above 4.2: "... then the error term becomes zero, and Algorithm 1 converges linearly to the fixed point solution ...". Which error rate becomes zero? be more specific please. Also, you never pointed out with respect to which parameter the rate is linear. Is it always \alpha?
5.  Paragraph above 4.2, last two sentences: the text seems to be messed up there, please fix.
6. Lemma 1: can you specify what exactly is C_0 or how can we estimate/bound it given some MDP? It is a caveat in your main and final result.

**Experience Assessment:**

I have published in this field for several years.

**Review Assessment: Checking Correctness Of Derivations And Theory:**

I assessed the sensibility of the derivations and theory.

**Review Assessment: Checking Correctness Of Experiments:**

I assessed the sensibility of the experiments.

**Review Assessment: Thoroughness In Paper Reading:**

I read the paper at least twice and used my best judgement in assessing the paper.

---

> ### Author Response · Authors · 2019-11-14
> **We thank the reviewer very much for the very helpful comments! Below is a point-to-point response.**
>
> Q1: Despite the advantages above, I am disturbed by the lack of comparison of the computational burden introduced by the M inner loops to vanilla TD. It is not clear from the results whether a practitioner would prefer paying those extra computations to reduce the bias and variance by the M-dependent factors in the convergence rate.
>
> A: Thank you so much for the comments. In the updated paper, we have added an additional section in supplemental materials (see Appendix E) to formally compare between VRTD and vanilla TD theoretically, in terms of the overall computational complexity (i.e., the total number of gradient computations) to achieve a common convergence accuracy. Our results show that VRTD outperforms TD under Markov sampling, and it also outperforms TD  under i.i.d. sampling for the case with a small conditional number. The reviewer can refer to the section for further details.
>
> As a side note, we were able to improve our bounds in Lemma 1 and Theorem 2, and the results in the revision reflect such changes.
>
>
> Q2: Second, the novelty of this work is not highly significant. It indeed corrects a bound for an existing algorithm, but not more than that. Namely, it only analyzes the so-called VRTD with a specific stepsize O(1/n), which also contains a system-dependent parameter that is not known to the user -- the smallest eigenvalue of the driving matrix.
>
> A: We find that VRTD is an important algorithm and hence deserves a correct theoretical analysis. In the technical level, our analysis of the bias error under Markovian sampling takes a different path from how the existing analysis of TD handles Markovian samples. Our proof captures explicitly how the variance reduction scheme helps to further reduce the bias error. Such an analysis further captures the advantage of VRTD over vanilla TD by a factor of log(1/\epsilon) reduction in the overall computational complexity, which is the new result that we added suggested by the reviewer. Such a result would not be possible if we follow the existing way of handling the Markovian samples.
>
> The stepsize issue mentioned by the reviewer is indeed a practical issue that requires further exploration. In our current experiments, we found that the convergence is not very sensitive to the stepsize when we set it to be 0.1.
>
>
> Further Questions:
>
> Q3:  In several locations, the TD update is referred to as a gradient. Despite the explanation as of why in Footnote 1, this is misleading. Since it is known that the TD update is *not* a true gradient step, it would be wiser not to refer to it as one.
>
> A: We agree and will be happy to make the change. Since such a change is throughout the paper, we would like to do it at the end of the review process in order not to cause any confusion.
>
>
> Q4: Can't Algorithm 1 be easily improved to be more efficient? In step 7 you perform M updates but end up throwing away M-t of them, where t is drawn randomly (step 10). Instead, you can simply draw t at the beginning and compute step 7 only up to that t. Is that correct?
>
> A: We thank the reviewer for providing this suggestion. Yes, we can randomly select an index t_m at m-th epoch and only update up to t_m-th iteration to make \theta_{t_m} as the output for the next epoch.
>
>
> Q5: Theorem 1: mention what exactly is the M you refer to -- that might expose more interesting relations.
>
> A:  In Theorems 1 and 2, we have provided the explicit expression for M in the revision
>
>
> Q6:  Paragraph above 4.2: "... then the error term becomes zero, and Algorithm 1 converges linearly to the fixed point solution ...". Which error rate becomes zero? be more specific please. Also, you never pointed out with respect to which parameter the rate is linear. Is it always \alpha?
>
> A: We have revised the sentence as suggested in our revision. The error ||\theta_m-\theta^*||^2_2 converges to zero at a linear rate as the iteration number m goes to infinity. The rate is linear with respect to the conditional number C_1, which is a positive constant and less than 1.
>
>
> Q7: Paragraph above 4.2, last two sentences: the text seems to be messed up there, please fix.
>
> A: Thanks for pointing this out. We have rewritten the two sentences in the revision.
>
>
> Q8: Lemma 1: can you specify what exactly is C_0 or how can we estimate/bound it given some MDP? It is a caveat in your main and final result.
>
> A: C_0 depends proportionally on the coupling time and the returning time of the underlying Markov chain. Specifically, the coupling time indicates how fast the Markov chain converges to the stationary distribution, and the returning time, which depends on the nature of the stationary distribution, captures the expected time that the Markov chain returns to the same state. The value of C_0 can be estimated based on the mixing time and the stationary distribution of the Markov chain.

---

### Official Review · AnonReviewer4 · 2019-11-04
**Official Blind Review #4**

**Rating:** 8

**Review:**

This paper presents a non-asymptotic analysis of Variance Reduced TD (VRTD), proposed by Korda and La (2015), to apply variance reduction ideas to temporal difference learning, specifically TD(0) with linear function approximation. The algorithm closely follows ideas of stochastic variance reduction (SVRG) which is widely used for large scale empirical risk minimization.

Specifically, the authors show VRTD converges (in expectation) to a neighborhood of the limit point using a constant step-size. Interestingly, this neighborhood can be made small using a large batch size M in both the IID and Markovian sampling regimes. From a technical standpoint, the work seems novel with interesting results although I am not sure if VRTD offers practical performance gains.

Main points:

1) I am a bit confused by the presence of a constant variance error term in the results. Intuitively, with variance reduction, I was expecting convergence in expectation; not just to a neighborhood with constant error. In other words, the variance error term decaying with $m$ (and not M). This would imply a stronger result than what authors have currently.

Note that vanilla SGD for strongly convex objectives also suffers from a constant variance error term (see for example Chapter 4 in Bottou et al., 2018). However, analysis of SVRG (Johnson and Zhang 2013) shows linear convergence for strongly convex objectives and claims sublinear convergence for convex objectives. While I appreciate the technical challenge in analyzing a \textit{semi-gradient} method like TD vis-a-vis  analyzing gradient descent for convex objectives, my understanding is that (Bhandari et al., 2018) showed a connection between the two. This connection makes me think if variance reduction can also help \textit{get rid} of the constant variance error term when analyzing TD(0) with constant step-sizes. Can the authors clarify?

Also, the authors might find it useful to look at (Lakshminarayanan and Szepesvari, 2018) which does in fact show convergence with a constant (problem instance independent) step-size for the IID case. I think that result only applies with iterate averaging but it might still be useful (certainly as a citation suggestion).

2) As a practical proposal, I wonder if VRTD has a potential benefit over vanilla TD(0). Essentially, the rates showed in this paper require $O(mM)$ samples. I wonder how vanilla TD would perform with $O(mM)$ samples coupled with a simple strategy of reducing the step-size by half when the value-function estimates stop changing. See Chapter 4 in (Bottou et al., 2018) for details. That would probably be a fairer comparison to help convince audience of VRTD as a practical alternative to TD(0).

Minor points:
- The counter example in Section 3.2 seems out of place. While it is important to point out the errors in previous analysis, I suggest the authors to flesh out the details in an Appendix section.
- $R_{\theta}$ and $r_{max}$ seem undefined in the main body of the paper.
- The constants in Theorem 1 and 2 seem complicated. Is there a way to simplify these for presentation purposes?

Response to author comments.

I thank the authors for their comments and changes to the draft.

1) The explanation makes a lot of sense. Since we cannot exactly estimate the population gradient, the variance error seems unavoidable with constant stepsize. I like the clarifications in Section 3.2

2) Thanks for clarifying that all the comparisons are done in terms of a fixed number of total gradient computations. Please add this clearly in the paper as well. I think the additional experiments and comparison with the new scheme I suggested above does a good job of convincing the readers of the practical usefulness of variance reduction.

Although I am not sure how likely practitioners are to use this (both $\alpha$ and M are problem dependent), in my opinion the paper presents a novel analysis of an important idea of variance reduction applied to Online TD. The experiments seem reasonable to suggest benefits. I’ll be happy to see this paper accepted. I have also updated my score.

Some minor comments and suggestions:

1) In Theorem 1, constant D_2 depends on $R_{\theta}$ but there Algorithm 1 has no projection step. I think the authors assume that $\norm{\theta^*} \leq R_{\theta}$. If so, please state that somewhere. If not, maybe I missed something and would like a clarification.

2) Page 6 should have $e(\tilde{theta}_m-1)$ instead of $e(\tilde{theta}_m)$.

3) Before stating the main results in Section 4, please restate the definition of $\lambda_A$ for better readability.

4) Constant G seems to be undefined for Theorem 2. In fact, it seems that $C_4 = G^2 [1 + \frac{2\rho \kappa}{1-\rho}]$. This is what I meant by simplifying constants. Ensuring that these constants are optimized for presentation helps.

5) Theorem 2 has a dependence on $d$. Maybe the authors should explicitly mention and discuss that.

6) The result for Markovian case requires the bias term to not dominate. This seems novel to me and would benefit from emphasis in the main body of the paper.

7) Equation 5 should have ’m’ instead of ’n’.

8) I always find large figures and clear (and) large axis labels very useful. Can the ‘iteration process’ graphs in Figure 1 be made more reader friendly? I’d  like to see the behavior of vanilla TD more clearly.

9) Section A in the Appendix might benefit from more background. For example, what is $v$. I am thinking this section to be more self contained with the readers not being compelled to read the proof in Korda and La (2015).


**Experience Assessment:**

I have published one or two papers in this area.

**Review Assessment: Checking Correctness Of Derivations And Theory:**

I assessed the sensibility of the derivations and theory.

**Review Assessment: Checking Correctness Of Experiments:**

I assessed the sensibility of the experiments.

**Review Assessment: Thoroughness In Paper Reading:**

I read the paper at least twice and used my best judgement in assessing the paper.

---

> ### Author Response · Authors · 2019-11-14
> **We thank the reviewer very much for the very helpful comments! Below is a point-to-point response.**
>
> Response to Question 1:
>
> Very good question! The traditional SVRG minimizes a finite-sum objective function, and can hence compute an exact full gradient in each loop for constructing the variance-reduced stochastic gradients. In this way, its variance can vanish upon convergence. However, for TD type of algorithms, the target “gradient” (which more rigorously should be referred to as Bellman operator update) is a population quantity in the form of expectation, not a form of finite-sum. Since VRTD can only sample a large batch of samples in each loop as an estimation of such a population gradient, a non-vanishing variance is inevitable, which leads to the constant error term in the final bound.
>
> We also thank the reviewer for pointing out an interesting and related paper, and we will cite and discuss it.
>
>
> Response to Question 2:
>
> To demonstrate the practical performance of VRTD, in the updated paper, we have provided additional experiments in Supplementary Materials Appendix G. More specifically, we compare VRTD with vanilla TD on two problems in OpenAI Gym, which are Frozen Lake and Mountain Car. Both experiments indicate that VRTD achieves a much smaller error than vanilla TD (i.e., M=1), and increasing the batch size for VRTD substantially reduces the error without much slowing down the convergence. We note that in our experiments, all algorithms are compared with respect to a common basis, i.e., they use the same total number of gradient computations (equivalently, they use the same total number of samples).
>
> Thank you for suggesting the TD scheme with a decreasing step-size, which we find to be interesting. In Appendix G3, we added an experiment on the Frozen Lake problem to compare such a TD scheme with VRTD. Our experiment demonstrates that VRTD still has a better performance than such a TD scheme.
>
> Although the reviewer’s question is from the practical side, we hope the reviewer will also find the following relevant. In the updated paper, we have added an additional section in supplemental materials (see Appendix E) to formally compare between VRTD and vanilla TD theoretically, in terms of the overall computational complexity (i.e., the total number of gradient computations) to achieve a common convergence accuracy. Our results show that VRTD outperforms TD under Markov sampling, and it also outperforms TD  under i.i.d. sampling for the case with a small conditional number. The reviewer can refer to the section for further details.
>
>
> Minor points:
>
> Q3: The counter example in Section 3.2 seems out of place. While it is important to point out the errors in previous analysis, I suggest the authors to flesh out the details in an Appendix section.
>
> A: In the revision, we moved the counter example to Appendix A.
>
>
> Q4: R_\theta and  r_max seem undefined in the main body of the paper.
>
> A: R_\theta is the projection radius and r_max is the upper bound of the reward. We have added these in the revision.
>
>
> Q5: The constants in Theorem 1 and 2 seem complicated. Is there a way to simplify these for presentation purposes?
>
> A: In the revision, we have provided simpler forms for the convergence results in Theorems 1 and 2 right below the two theorems.

---

### Decision · Program_Chairs · 2019-12-19

**Decision:**

Accept (Poster)

**Comment:**

The paper studies the variance reduced TD algorithm by Konda and Prashanth (2015). The original paper provided a convergence analysis that had some technical issues. This paper provides a new convergence analysis, and shows the advantage of VRTD to vanilla TD in terms of reducing the bias and variance. Several of the five reviewers are expert in this area and all of them are positive about it. Therefore, I recommend acceptance of this work.